



# Overview: Quasi-Lagrangian observations of Arctic air mass transformations – Introduction and initial results of the HALO–$(\mathcal{AC})^3$ aircraft campaign

Manfred Wendisch[1], Susanne Crewell[2], André Ehrlich[1], Andreas Herber[3], Benjamin Kirbus[1], Christof Lüpkes[3], Mario Mech[2], Steven J. Abel[4], Elisa F. Akansu[5], Felix Ament[6], Clémantyne Aubry[7,8], Sebastian Becker[1], Stephan Borrmann[9,10], Heiko Bozem[10], Marlen Brückner[1], Hans-Christian Clemen[9], Sandro Dahlke[3], Georgios Dekoutsidis[7], Julien Delanoë[8], Elena De La Torre Castro[7,10,11], Henning Dorff[6], Regis Dupuy[12], Oliver Eppers[9], Florian Ewald[7], Geet George[13,14], Irina V. Gorodetskaya[15], Sarah Grawe[5], Silke Groß[7], Jörg Hartmann[3], Silvia Henning[5], Lutz Hirsch[13], Evelyn Jäkel[1], Philipp Joppe[9,10], Olivier Jourdan[12], Zsofia Jurányi[3], Michail Karalis[16], Mona Kellermann[5], Marcus Klingebiel[1], Michael Lonardi[1,17], Johannes Lucke[7,11], Anna Luebke[1], Maximilian Maahn[1], Nina Maherndl[1], Marion Maturilli[3], Bernhard Mayer[18], Johanna Mayer[7], Stephan Mertes[5], Janosch Michaelis[3,19], Michel Michalkov[5], Guillaume Mioche[12], Manuel Moser[7,10], Hanno Müller[1], Roel Neggers[2], Davide Ori[2], Daria Paul[2], Fiona Paulus[2], Christian Pilz[5], Felix Pithan[3], Mira Pöhlker[5], Veronika Pörtge[18], Maximilian Ringel[6], Nils Risse[2], Gregory C. Roberts[20], Sophie Rosenburg[1], Johannes Röttenbacher[1], Janna Rückert[21], Michael Schäfer[1], Jonas Schaefer[5], Vera Schemann[2], Imke Schirmacher[2], Jörg Schmidt[1], Sebastian Schmidt[22], Johannes Schneider[9], Sabrina Schnitt[2], Anja Schwarz[1], Holger Siebert[5], Harald Sodemann[23,24], Tim Sperzel[1], Gunnar Spreen[21], Bjorn Stevens[13], Frank Stratmann[5], Gunilla Svensson[16], Christian Tatzelt[5], Thomas Tuch[5], Timo Vihma[25], Christiane Voigt[7,10], Lea Volkmer[18], Andreas Walbröl[2], Anna Weber[18], Birgit Wehner[5], Bruno Wetzel[5], Martin Wirth[7], and Tobias Zinner[18]

[1]Leipziger Institut für Meteorologie (LIM), Universität Leipzig, Leipzig, Deutschland
[2]Institut für Geophysik und Meteorologie (IGM), Universität zu Köln, Köln, Deutschland
[3]Alfred–Wegener–Institut, Helmholtz–Zentrum für Polar– und Meeresforschung (AWI), Bremerhaven & Potsdam, Deutschland
[4]Met Office, Exeter, United Kingdom
[5]Leibniz–Institut für Troposphärenforschung (TROPOS), Leipzig, Deutschland
[6]Meteorologisches Institut, Universität Hamburg, Hamburg, Germany
[7]Institut für Physik der Atmosphäre, Deutsches Zentrum für Luft- und Raumfahrt (DLR), Oberpfaffenhofen, Deutschland
[8]Laboratoire Atmosphères, Milieux et Observations Spatiales (LATMOS), Centre National de la Recherche Scientifique (CNRS), Guyancourt, France
[9]Abteilung für Partikelchemie, Max-Planck-Institut für Chemie (MPIC), Mainz, Deutschland
[10]Institut für Physik der Atmosphäre (IPA), Johannes Gutenberg-Universität, Mainz, Deutschland
[11]Faculteit Luchtvaart- en Ruimtevaarttechniek, Technische Universiteit Delft, Delft, Nederland
[12]Laboratoire de Météorologie Physique (LaMP), Université Clermont Auvergne, Centre National de la Recherche Scientifique (CNRS), Clermont-Ferrand, France
[13]Max-Planck-Institut für Meteorologie (MPIM), Hamburg, Deutschland
[14]now at: Delft University of Technology (TU Delft), Delft, Netherlands
[15]Centro de Estudos do Ambiente e do Mar (CESAM), Universidade de Aveiro, Aveiro, Portugal
[16]Department of Meteorology and Bolin Centre for Climate research, Stockholm University, Stockholm, Sweden



[17]Now at: Extreme Environments Research Laboratory (EERL), Ecole Polytechnique Fédérale de Lausanne (EPFL), Sion, Switzerland

[18]Meteorologisches Institut, Ludwig-Maximilians-Universität München, München, Deutschland

[19]Now at: Maritime Klimatologie, Maritim-klimatologische Analysen und Produkte, Deutscher Wetterdienst (DWD), Hamburg, Deutschland

[20]Scripps Institution of Oceanography, University of California San Diego, La Jolla, USA

[21]Institut für Umweltphysik (IUP), Universität Bremen, Bremen, Deutschland

[22]Department of Atmospheric and Oceanic Sciences, Laboratory for Atmospheric and Space Physics (LASP), University of Colorado, Boulder, USA

[23]Geophysical Institute, University of Bergen, Bergen, Norway

[24]Bjerknes Centre for Climate Research, Bergen, Norway

[25]Finnish Meteorological Institute (FMI), Helsinki, Finland

**Correspondence:** Manfred Wendisch (m.wendisch@uni-leipzig.de)

**Abstract.** The global warming is amplified in the Arctic. To collect data that help to constrain weather and climate models, which often do not realistically represent the enhanced Arctic warming, the HALO–$(\mathcal{AC})^3$ aircraft campaign was conducted in March and April 2022 over the Norwegian and Greenland Seas, the Fram Strait, and the central Arctic Ocean. Observations were made over areas of open ocean, the marginal sea ice zone, and the central Arctic sea ice. Two low-flying and one long-range, high-altitude research aircraft have been employed. Whenever possible, the three aircraft were flown in collocated formation. The campaign focused on one specific challenge posed by the models: The reasonable representation of transformations of air masses during their meridional transport into (northward by moist and warm air intrusions, WAIs) and out of (southward via marine cold air outbreaks, CAOs) the Arctic. To observe the air mass transformations, a quasi-Lagrangian flight strategy using trajectory calculations was realized enabling to sample the moving air mass parcels twice along their trajectories. Eight distinct WAI and 12 CAO cases were probed extensively. From the quasi-Lagrangian measurements, we have derived the diabatic heating and moistening of the moving air masses during CAOs and WAIs, the development of cloud macrophysical and microphysical properties along the southward pathways of the air masses during CAOs, and the moisture budget of WAIs. As an example result, we have obtained typical values of the surface-driven diabatic heating between 1–3 K h$^{-1}$ and of the near-surface moistening between 0.05–0.3 g kg$^{-1}$ h$^{-1}$ within the lowest about 0.5 km. From the observations of WAIs, a weak diabatic cooling of up to 0.4 K h$^{-1}$ and a moisture loss of up to 0.1 g kg$^{-1}$ h$^{-1}$ from the ground to about 5 km altitude were derived. In addition, we discuss the frequency of occurrence of the different thermodynamic phases of Arctic low-level clouds, the interaction of Arctic cirrus with sea ice, water vapor, and aerosol particles, and the characteristic microphysical and chemical properties of Arctic aerosol particles. Finally, we provide proof of concept to measure mesoscale divergence and subsidence in the Arctic using data from dropsondes released during circular flight patterns.

# 1 Introduction

In 2017, anthropogenic global warming reached around 1 K above pre-industrial levels (Masson-Delmotte et al., 2021). To date, human-caused climate change has resulted in 1.2 K of warming (Forster et al., 2023). Data published by the Copernicus Climate Change Service show that in 2023, on some days, the daily global average temperature was more than 1.5 K warmer



than during pre-industrial times. The advancing global warming triggers numerous feedback mechanisms in the Earth's climate
system, most of which are not fully accounted for in climate models (Ripple et al., 2023). Of the 41 important feedback loops
identified by Ripple et al. (2023), at least a quarter cause relevant, mainly amplifying effects in the Arctic. This makes the
Arctic one of the "hot spots" of global climate change (Overland et al., 2011).

As a result of global warming and these interlinked feedback mechanisms, a new Arctic has emerged over the last three
decades (Jeffries et al., 2013). The rapidly progressing changes in many Arctic climate parameters are obviously imprinted in
properties such as Arctic sea ice (extent, thickness, age), sea surface temperature, near-surface air temperature, Greenland ice
sheet and glaciers, terrestrial snow cover, and permafrost, precipitation, primary biological productivity of the Arctic Ocean
and others (Druckenmiller et al., 2022). This enhanced response of the Arctic climate system to global warming is commonly
referred to as Arctic amplification (Serreze and Francis, 2006; Graversen et al., 2008; Serreze et al., 2009; Serreze and Barry,
2011; Previdi et al., 2021).

One important indication of Arctic amplification is the up to four times faster increase in Arctic near-surface air temperature
compared to global warming over the last three to four decades, which fits only poorly into the scatter of the multi-model
ensemble results of the Coupled Model Intercomparison Project (CMIP), Phase 5 (CMIP5) and Phase 6 (CMIP6) (Holland and
Landrum, 2021; Rantanen et al., 2022). Chylek et al. (2022) concludes that after 1999, Arctic amplification appears to be caused
by internal variability, which is not appropriately represented in CMIP6 models. Further obvious signs of Arctic amplification
are the faster-than-expected remarkable decline of sea ice cover of the Arctic Ocean since around 1970, especially in late
summer (Stroeve et al., 2007; Olonscheck et al., 2019; Serreze and Meier, 2019; Screen, 2021), and the gradual thawing of the
permafrost soils (Beer et al., 2020). These changes have important consequences for the living conditions of the local Arctic
human population, as well as for the flora and fauna of the Arctic. They also imply potentially far-reaching economic impacts
for fishing in Arctic waters, transoceanic shipping routes, tourism, and the extraction of natural resources.

Arctic amplification has long been understood to be a feature of global climate change (Manabe and Wetherald, 1975).
More recently, knowledge and understanding of the processes and feedback mechanisms governing Arctic amplification have
improved considerably (Previdi et al., 2021; Smith et al., 2021; Taylor et al., 2022; Wendisch et al., 2023a). Nevertheless, the
current ability to model them is still limited and, therefore, future model-based projections of Arctic climate changes are highly
uncertain (Smith et al., 2019; Cohen et al., 2020; Block et al., 2020; Linke et al., 2023). In particular, the model representations
of the effects and development of clouds (Pithan et al., 2014; Wendisch et al., 2019; Kretzschmar et al., 2020; Stevens and Kluft,
2023), and of the interactions of the atmosphere with sea ice, snow on sea ice, and ocean physics as well as biogeochemical
feedback processes are challenging (Rinke et al., 2019; Huang et al., 2019; Pefanis et al., 2020). In addition, the role of aerosol
particles in Arctic amplification has not been sufficiently investigated (Schmale et al., 2021; Dada et al., 2022; Gong et al.,
2023).

For more than a decade, there is some debate as to whether climate changes in the Arctic will impact the weather and climate
in the mid-latitudes (Cohen et al., 2014). Several dynamic processes in the lower and upper troposphere and the stratosphere
may lead to a weaker or stronger jet stream (Francis and Vavrus, 2015; Blackport and Screen, 2020; Yuval and Kaspi, 2020),
with consequences on the meandering (amplitude) and persistence of the Rossby waves. These dynamical effects influence



the meridional transport of heat, moisture, and momentum through northward warm air intrusions (WAIs[1]), including so-
called atmospheric rivers (ARs), and southward cold air outbreaks (CAOs[2]). More frequent WAIs could further enhance Arctic
warming (Pithan et al., 2018; Nash et al., 2018), whereas CAOs may contribute to cold events in mid-latitudes. Recent studies
suggest that despite of the Arctic warming, cold winter events in mid-latitudes have remained nearly as extreme and as common
as decades ago (Cohen et al., 2023; Nygård et al., 2023). However, the results of the Polar Amplification Model Intercomparison
Project (PAMIP) show very little evidence of linkages (Smith et al., 2022).

In any case, Rossby waves realize the meridional transport of air masses into and out of the Arctic. It is estimated that WAIs
increase total column water vapor and cloud prevalence in the Arctic winter by about 70 % and 30 %, respectively (Johansson
et al., 2017). As a result, stronger thermal-infrared downward radiation reduces the net surface radiative cooling and increase
the near-surface air temperature in winter by about 5 K (Johansson et al., 2017). This warming could trigger an earlier onset of
melting and more melt ponds, which would reduce the surface albedo. In addition, particles and pollution are transported into
the Arctic during WAIs, which may influence cloud properties (Bossioli et al., 2021).

In spite of the high impact they have on the Arctic climate, there are problems in modeling air mass transformations during
meridional transport (Sato et al., 2016; Pithan et al., 2016; Dimitrelos et al., 2020). In particular, large-scale models have
difficulties representing important thermodynamic processes driving Arctic air mass transformations, including the evolution
of microphysical properties of mixed-phase clouds (McCoy et al., 2015; Pithan et al., 2014; Tan and Storelvmo, 2019), the
development of turbulent fluxes under stable stratification (Tjernström et al., 2005; Holtslag et al., 2013; Gryanik and Lüpkes,
2023), and the response of snow-covered sea ice to atmospheric forcing (Pithan et al., 2023). These processes control the
response of the Arctic to climatic forcing, and their realistic representation in models is crucial for understanding the behavior
and feedback mechanisms of the Arctic climate system (Block et al., 2020; Taylor et al., 2022).

While the large-scale conditions that favor the development of CAOs can be well predicted on sub-seasonal time scales, a
better understanding of when and how CAOs lead to the development of polar lows is necessary to predict these events, which
often have large impacts (Polkova et al., 2021). Furthermore, improvements to the observing system and the understanding
and model representation of small-scale synoptic features and processes are needed to advance polar predictions on daily
to seasonal time scales (Jung et al., 2016). Moreover, given the lack of in-situ observations, models are often evaluated using
reanalysis data. However, the fidelity of such atmospheric reanalyses can suffer from a low number of assimilated observations,
as well as biases inherited from its driving model (Tjernström and Graversen, 2009).

As a consequence, dedicated quasi-Lagrangian observations of WAIs and CAOs would be helpful to improve the model
capabilities to realistically represent processes that determine air mass transformations during meridional transport into and
out of the Arctic (Wendisch et al., 2021). Previous observations of air mass transformations in the Arctic have mostly been
conducted in the framework of case studies at locally fixed, ground-based positions (Eulerian point of view) partly combined
with ship, aircraft, or satellite data. Examples include case studies based on data from the Surface Heat Budget of the Arctic
Ocean (SHEBA) (Uttal et al., 2002) and the Multidisciplinary drifting Observatory for the Study of Arctic Climate (MOSAiC)

---

[1]In our perspective, WAIs include the northward transport of both warm and humid air.

[2]In this paper we restrict ourselves to marine CAOs.



(Shupe et al., 2022; Kirbus et al., 2023a; Svensson et al., 2023) ship expeditions. Due to the lack of geostationary satellite data, the development of Arctic cloud properties can only be analyzed by polar orbiting satellite observations from an Eulerian perspective assuming stationary conditions (Murray-Watson et al., 2023). However, the Eulerian approach does not permit the required observations of temporal air mass-transforming processes.

Only very few quasi-Lagrangian aircraft-based studies have been carried out so far (Boettcher et al., 2021), and none in the Arctic. Furthermore, several studies use atmospheric reanalysis to identify and discuss WAIs or CAOs (You et al., 2021b; Kirbus et al., 2023b). Some authors have put local icebreaker and airborne or satellite observations into a Lagrangian framework by means of trajectories based on reanalysis (Tjernström et al., 2019; Ali and Pithan, 2020; You et al., 2021a; Kirbus et al., 2023a; Mateling et al., 2023). Another approach was performed in earlier aircraft-based campaigns with measurements along the mean wind direction in CAO conditions and WAIs (Hartmann et al., 1997; Brümmer and Thiemann, 2002; Vihma et al., 2003; Lüpkes et al., 2012; Chechin et al., 2013). These authors have focused on the local development around the Fram Strait and the close marginal sea ice zone (MIZ) with measurements above sea ice and open ocean. A disadvantage was that the quasi-Lagrangian analysis had to assume stationary conditions.

Therefore, we have conducted the HALO–$(\mathcal{AC})^3$ aircraft campaign (HALO: High Altitude and Long Range Research Aircraft, $(\mathcal{AC})^3$: Project on Arctic Amplification: Climate Relevant Atmospheric and Surface Processes, and Feedback Mechanisms, see https://halo-ac3.de/). Based on the open issues in modeling air mass transformations during meridional transport into and out of the Arctic, the HALO–$(\mathcal{AC})^3$ mission pursued two general objectives (Wendisch et al., 2021). The first was to jointly use HALO and the Polar 5 (P5) and Polar 6 (P6) research aircraft to perform quasi-Lagrangian observations of air mass transformations during WAIs and CAOs – an approach that has not been tried before in the Arctic. The second was to test the ability of numerical atmospheric models to reproduce the measurements taken from the aircraft. The benchmarked models can then for example be applied to investigate linkages between Arctic amplification and mid-latitude weather. This paper describes efforts carried out in support of the first objective.

The article is structured in five sections. After the introduction, the mainly utilized three research aircraft and their instrumentation, as well as the collocated flights are described in Sect. 2. The unique quasi-Lagrangian measurement strategy successfully applied during the campaign is presented in Sect. 3. Some initial results from HALO–$(\mathcal{AC})^3$ and ongoing analysis are discussed in Sect. 4. The following themes are elaborated: Air mass transformations during WAIs and CAOs (Sect. 4.1), Arctic clouds (Sect. 4.2) and aerosol particles (Sect. 4.3), and a proof of concept to measure mesoscale divergence and subsidence in the Arctic (Sect. 4.4). A summary and some conclusions are given in Sect. 5. Three appendices cover some interesting supplementary aspects of the HALO–$(\mathcal{AC})^3$ aircraft campaign.

## 2 Aircraft, instrumentation, and flight pattern

Three research aircraft were mainly involved in the HALO–$(\mathcal{AC})^3$ aircraft campaign: HALO (High Altitude and Long Range Research Aircraft), Polar 5 (P5), and Polar 6 (P6). HALO is operated by the German Aerospace Center (Deutsches Zentrum für Luft- und Raumfahrt, DLR). It was based in Kiruna (northern Sweden). The P5 and P6 were stationed at Longyearbyen (Sval-



bard, Norway). These two aircraft belong to the Alfred Wegener Institute, Helmholtz Center for Polar and Marine Research (AWI). In addition, the British Facility for Airborne Atmospheric Measurements (FAAM) and the French Avions de Transport Régional (ATR) aircraft were concurrently based in Kiruna measuring partly in coordination with HALO, P5, and P6. The ATR was operating from 22 March to 3 April 2022, and had some similar objectives as HALO, but with the addition of using in-situ water isotope measurements to characterize processing of water vapor in these weather systems. The FAAM aircraft

operated from 7 March to 1 April 2022, and focused on making in-situ measurements of the development of CAOs from the sea ice around Svalbard to the Scandinavian coastline. FAAM was fitted with a range of instrumentation to measure the thermodynamic, aerosol, cloud, and precipitation properties within CAOs. An overview of the instrumentation and all data collected during the campaign period is given by Ehrlich et al. (2024). Furthermore, intensive ground-based measurements were carried out at the AWIPEV research base in Ny-Ålesund, including additional observations with a tethered balloon (Lonardi et al.,

135 2024).

In the following, the three main aircraft used during HALO–$(\mathcal{AC})^3$ are described. HALO – a Gulfstream G550 – has sufficient range/endurance (up to 9000 km and 10 hours) for quasi-Lagrangian air mass observations. HALO is capable to lift up to three tons of state-of-the-art meteorological and remote sensing instruments up to 15 km altitude to observe the complete vertical tropospheric air mass column, including water vapor, aerosol particles, clouds, precipitation and surface properties.

HALO was equipped with a unique remote sensing payload that has matured in several campaigns in the past (Stevens et al., 2019; Konow et al., 2021). The instrumentation includes a 26-channel microwave radiometer, a 35 GHz Doppler radar, aerosol and water vapor lidar, spectral and broadband solar and thermal-infrared radiation sensors, which are upward and downward looking, imaging and polarization camera spectrometers in the solar and thermal-infrared spectral ranges, and dropsondes. HALO was operated from Kiruna between 7 March and 12 April 2022. 17 research flights (RFs) were conducted (RF02–

RF18) with a total flight time of 147 hours (Fig. 1a). In total, 330 dropsondes were released during the RFs from HALO (Tab. 1).

The low-flying P5 and P6 (Wesche et al., 2016) aircraft are Basler BT-67 (DC-3) types with a ceiling of up to 6 km, and a range of about 2300 km. Each of the two planes can carry a scientific payload that can weigh up to 1 ton. P5 provided active and passive remote sensing measurements to characterize clouds, precipitation, aerosol particles, trace gases, and surface properties

from atop, similar to HALO. The instrumentation of P5 included a 94 GHz radar, an aerosol lidar, passive microwave, and radiation sensors. In addition, P5 carried in-situ instrumentation to derive turbulence parameters and energy fluxes (Mech et al., 2019; Schirmacher et al., 2023). Dropsondes were also released from P5.

P6 focused on in-situ measurements in the lower troposphere of below 2–4 km altitude in cloudy and cloud-free conditions. The measurements of P6 aimed to determine radiative and turbulent energy fluxes as well as to investigate smaller-scale

processes. For this purpose, P6 was equipped with in-situ probes to measure cloud and precipitation particles (droplets, ice crystals) and cloud residuals, aerosol particles, radiation, chemistry, and trace gas properties. For cloud observations, the P6 aircraft was equipped with a Cloud Droplet Probe (CDP), Cloud Imaging Probe (CIP), 2D-Stereo Probe (2D-S), and Precipitation Imaging Probe (PIP) for cloud particle counting and sizing, along with a Polar Nephelometer (PN) for measuring scattering properties and phase discrimination (Wendisch and Brenguier, 2013; Kirschler et al., 2023; De La Torre Castro et al.,



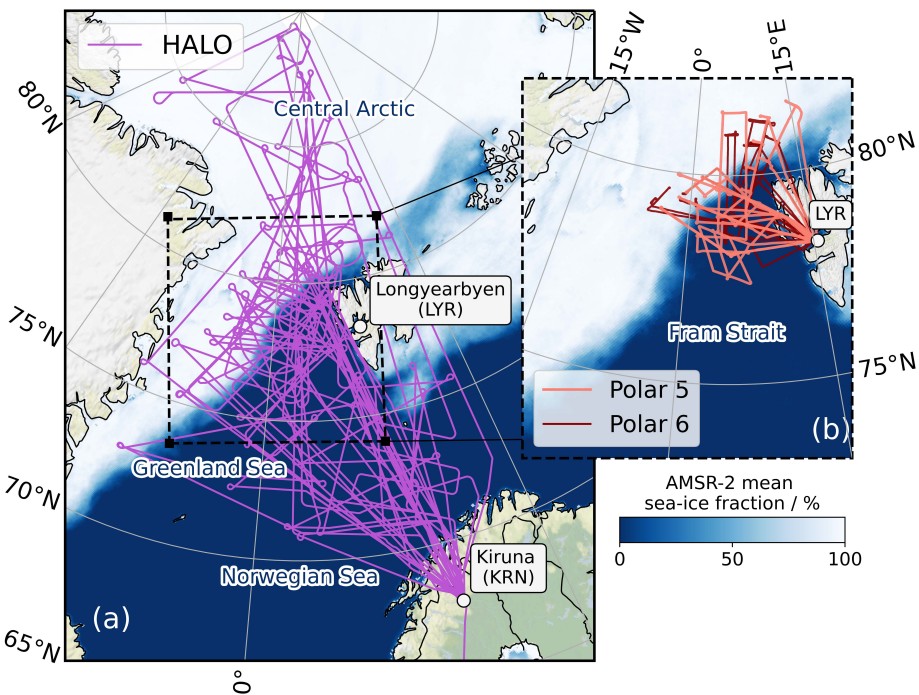

**Figure 1.** Flight paths (a) of HALO operating from Kiruna (KRN), and (b) of Polar 5 (P5) and Polar 6 (P6) aircraft based in Longyearbyen (LYR).

2023). The total data set of in-situ cloud measurements with P6 below 1.5 km sums up to about 21 hours, where about 15.5 hours were spent above the open ocean and 3.5 hours over sea ice.

The P5 and P6 aircraft followed a proven measurement strategy with vertically stacked, collocated remote sensing measurements above clouds (P5) and in-situ sampling inside clouds at lower altitudes (P6). Experiences with this measurement strategy were collected during the Arctic Cloud Observations Using airborne measurements during polar Day (ACLOUD) campaign performed in 2017 (Wendisch et al., 2019). During HALO–$(\mathcal{AC})^3$ both aircraft performed 13 RFs with 53 (P5) and 63 (P6) flight hours, respectively (Fig. 1b). 141 dropsondes were released during the flights of P5 (Tab. 1). Some coordinated flights between HALO and the P5 and P6 aircraft including closely vertically collocated flight segments were conducted. In addition, several joint flights of HALO, the ATR, and FAAM planes were realized.

# 3 Quasi-Lagrangian sampling

## 3.1 General approach

True Lagrangian observations would require flying embedded in a moving air mass parcel and recording data characterizing the changes of the air mass properties during its movement. The closest approach to achieving this would be balloons continuously



**Table 1.** Overview of HALO–$(\mathcal{AC})^3$ research flights (RFs) including the RF number and information on the coordination of HALO (High Altitude and Long Range Research Aircraft) with the P5 (Polar 5), P6 (Polar 6), FAAM (Facility for Airborne Atmospheric Measurements), and ATR (Avions de Transport Régional) aircraft. Furthermore, the synoptic situation (warm air intrusion, WAI, cold air outbreak, CAO, Arctic cirrus, AC, and polar low, PL) and overpasses of P5 over AWIPEV research base in Ny-Ålesund are indicated. The number of successfully launched dropsondes is given together with the number of dropsondes used in the Global Telecommunication System (GTS) data assimilation.

| | Research Flight (RF) Number | | | Coordination of | Synoptic | P5 over | Number of Dropsondes | | |
|---|---|---|---|---|---|---|---|---|---|
| | HALO | P5 | P6 | HALO with ... | Situation | AWIPEV | P5 | HALO | GTS |
| WARM & HUMID PERIOD | | | | | | | | | |
| DAY IN 2022 | | | | | | | | | |
| 12 March | 02 | – | – | | WAI | | – | 20 | – |
| 13 March | 03 | – | – | | WAI | | – | 21 | – |
| 14 March | 04 | – | – | | WAI | | – | 9 | – |
| 15 March | 05 | – | – | | WAI | | – | 25 | 3 |
| 16 March | 06 | – | – | FAAM | WAI | | – | 23 | 19 |
| 20 March | 07 | 01 | 01 | P5, P6 | WAI | YES | 12 | 17 | 13 |
| COLD & DRY PERIOD | | | | | | | | | |
| DAY IN 2022 | | | | | | | | | |
| 21 March | 08 | – | – | FAAM | CAO | | – | 13 | 13 |
| 22 March | – | 02,03 | 02 | | CAO | YES | 12 | – | – |
| 24 March | – | – | 03 | | CAO | | – | – | – |
| 25 March | – | 04 | – | | CAO | YES | 5 | – | – |
| 26 March | – | – | 04 | | CAO | | – | – | – |
| 28 March | 09 | 05 | 05 | P5, P6 | CAO | YES | 15 | 16 | 16 |
| 29 March | 10 | 06,07 | 06 | P5, P6, ATR, FAAM | CAO | YES | 5 | 18 | 10 |
| 30 March | 11 | 08 | 07 | P5, P6, ATR, FAAM | CAO | YES | 15 | 32 | 32 |
| 01 April | 12 | 09 | 08 | P5, P6 | CAO | YES | 18 | 41 | 41 |
| 04 April | 13 | 10 | 09 | P5, P6 | CAO | | 14 | 13 | 11 |
| 05 April | – | 11 | 10 | | CAO | | 10 | – | – |
| 07 April | 14 | 12 | – | P5 | AC | YES | 17 | 15 | 10 |
| 08 April | 15 | – | 11 | P6 | PL | | – | 21 | 5 |
| 09 April | – | – | 12 | | CAO/PL | | – | – | – |
| 10 April | 16 | 13 | 13 | P5, P6 | AC/WAI | | 18 | 22 | 21 |
| 11 April | 17 | – | – | | AC | | – | 7 | 6 |
| 12 April | 18 | – | – | | AC | | – | 17 | 16 |



accompanying the air mass and simultaneously drifting along with it. However, such balloon measurements also suffer some drawbacks (Businger et al., 1996; Johnson et al., 2000; Businger et al., 2006; Roberts et al., 2016). Because of their non-zero

inertia, drifting balloon observations do not pose a fully Lagrangian approach. In addition, due to vertical wind shear of wind speed and direction, the air mass trajectories follow different pathways in different altitudes. It quickly becomes a non-trivial task to select an altitude level to follow. Furthermore, the payload of such balloons is very limited.

Here, we avoid these drawbacks by following a new strategy termed quasi-Lagrangian and that accounts, e.g., for the height dependence of trajectories. Instead of relying on the limited payload capacity of balloons, we employ three different research

aircraft, equipped with instrumentation that has shown its potential in state-of-the-art atmospheric observations. Naturally, such aircraft move much faster than the relatively slow air masses moving with the respective wind speeds. To mitigate this problem, we design appropriate flight patterns aiming to sample air mass parcels along their pathway as often as possible. To realize this idea we utilize air mass trajectory calculations to determine the pathway of the air mass parcels. In case the air mass trajectory initially sampled is crossing the flight path at any level below a second time we define this as a quasi-Lagrangian match.

This approach is exemplified by the sketch presented in Fig. 2 for the case of a WAI. At a time $t_1$, we observe a vertical air Column 1 consisting of stacked air mass parcels. One such air mass parcel is indicated as a blue cube. A number of height-resolving remote sensing instruments aboard the high-flying HALO and P5, such as radar, lidar and microwave radiometer, and dropsondes characterize the properties of the air mass parcels at different altitudes at $t_1$. In addition, we directly probe the air mass parcels with the P6 aircraft collecting in-situ data. Besides many other quantities, dry potential air temperature ($\theta$),

specific humidity ($q$), air temperature ($T$), and radar reflectivity ($Z$) are measured by remote sensing and in-situ instruments.

In the next step, we use forward trajectories (dashed arrows in Fig. 2) to follow the pathway of the individual air mass parcels of the vertical air column. To calculate the trajectories, we define a horizontal circular area with a radius of 30 km in the center of the air mass parcel. As an example, we refer to the dashed ellipse within the blue cube of Column 1. Starting points are evenly spaced horizontally every 10 km, which results in about 30 regularly distributed points per starting altitude. From the

starting points, simulations of the 30 forward-trajectories per starting altitude are performed to project the average movement of the corresponding air mass parcel (for example the blue cube in Fig. 2). Subsequently, we follow the same approach for each air mass parcel within Column 1 to project the average movement of the air parcels as a function of altitude.

The vertical geometric thickness of the individual air mass parcels is assumed as 5 hPa, which also corresponds to the vertical resolution of the horizontal circular areas. The top of the vertical column is defined as 250 hPa (approximate flight

altitude of HALO), which results in a total of 150 (750 hPa divided by 5 hPa) air mass parcels and horizontal circular start areas for the trajectories. For each of the 150 air mass parcels in Column 1, 4500 trajectories are started (150 parcels × 30 regularly spaced initial points). These trajectories describe the movements of the air mass parcels in Column 1, which were sampled by the three aircraft. Now, the whole procedure is repeated along the entire flight track with a temporal resolution of one minute. In the case of HALO, the approximate flight time during the HALO–$(\mathcal{AC})^3$ campaign was about 8 hours per RF,

which means that $4500 \times 480 = 2.2 \times 10^6$ air mass parcel trajectories have been calculated for each HALO flight during the campaign.





Flight planning to realize quasi-Lagrangian observations required substantial efforts for the mission coordination. To combine flight tracks with the projected trajectories, the Mission Support Tool was used (Bauer et al., 2022).

### 3.2 Trajectories to find matches

The actual computation of the forward-trajectories of the air mass parcels was performed using the Lagrangian analysis tool (LAGRANTO) (Sprenger and Wernli, 2015). During the campaign, the trajectory calculations were based on the Integrated Forecast System (IFS) of the European Centre for Medium-Range Weather Forecasts (ECMWF) wind product. For processing the data after the campaign (for this paper) we have applied the Fifth Generation ECMWF Atmospheric Reanalysis (ERA5) (Hersbach et al., 2020). Both methods (IFS- and ERA5-based) have assimilated the dropsonde profile observations of ther-
modynamic and wind data taken during the flight (Table 1). Trajectories were calculated 60 hours forward in time. IFS and ERA5 were retrieved on 137 model levels, which are vertically spaced between the surface and top-of-atmosphere on a regular $0.25° × 0.25°$ latitude/longitude grid with a one-hourly temporal resolution. The improved performance of ERA5 compared to alternative atmospheric reanalyses data has been shown by Graham et al. (2019a, b).

Figure 2 illustrates the quasi-Lagrangian approach used for flight planning. The flight plans were prepared such that there
are enhanced chances to meet the air mass parcel indicated as a blue cube a second time along its trajectory during the flight. This case is illustrated in Column 2 where actually the same blue cube was encountered again by the aircraft at time $t_2$. Such a quasi-Lagrangian match is counted only (i) after at least 60 min of air mass parcel drift, and (ii) if the air mass parcel trajectory crosses the aircraft track within a 30 km radius. Each quasi-Lagrangian match enables to compare the air mass properties measured in Column 1 (subscript 1) with those measured during the quasi-Lagrangian match in Column 2 (subscript 2). Thus,
the temporal tendency of a thermodynamic or cloud property $\psi$ (e.g., dry potential air temperature $\theta$, specific humidity $q$, and air temperature $T$) that characterizes the rate of change of a particular air mass parcel along its trajectory can be quantified using the following equation:

$$\frac{\Delta\psi}{\Delta t} \quad = \quad \frac{\psi_2 - \psi_1}{t_2 - t_1}. \tag{1}$$

### 3.3 Example and statistics of matches

In the following, we illustrate the quasi-Lagrangian procedure for two consecutive HALO research flights (RF03 and RF04) performed within two days after each other (13 and 14 March 2022). In Fig. 3, the notable corridor of increased water vapor transport (IVT) indicates a substantial WAI event. During RF03 on 13 March, the HALO flight path (blue line) was designed to cover the WAI by a horizontal zig-zag flight pattern (Fig. 3a-b). HALO took off shortly after 8 UTC in Kiruna. The forward-trajectory simulations were started along the HALO flight path with a one-minute resolution (black lines), whereby only a
subset of simulated trajectories is depicted in the figure that later on during RF04 matched the HALO flight path a second time. These trajectories evolved until HALO landed in Kiruna around 17 UTC. During the night, the trajectories advanced indicating a continued transport of the humid air masses pole-ward (Fig. 3c). HALO took off on the second day (RF04) on 14 March around 10 UTC and crossed the trajectories that had been initiated the day before (Fig. 3d-e). Altogether, about 80,000



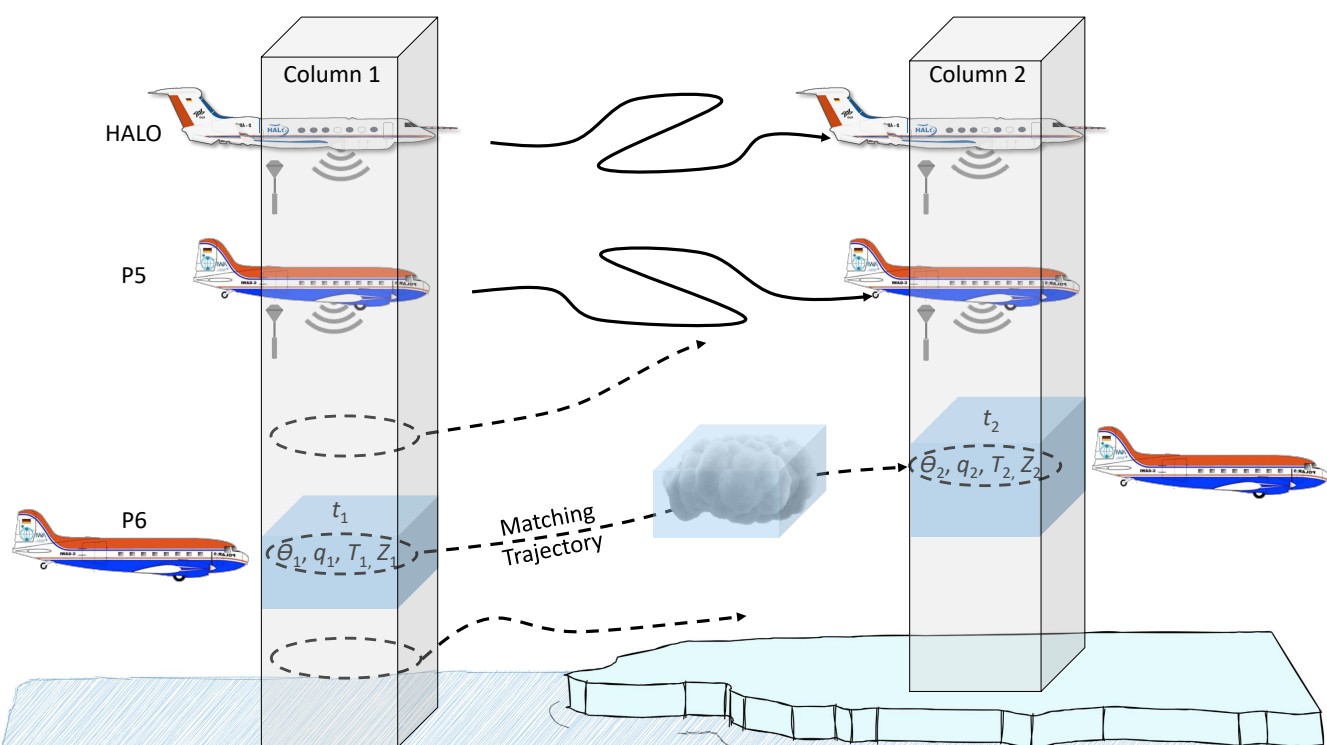

**Figure 2.** Illustration of the quasi-Lagrangian approach adopted during the HALO–$(\mathcal{AC})^3$ airborne campaign. At an initial time $t_1$ and within the atmospheric Column 1, air mass parcels (blue cubes) are observed using remote sensing instrumentation installed on HALO and P5. Trajectories are initiated that describe the movement of the air mass parcels. If they cross the flight path of the aircraft at a later time $t_2$, the air mass parcel can be sampled a second time within the atmospheric Column 2. This approach enables to observe the changes of the properties of the air mass parcel (for example, $\theta, q, T$, and $Z$ in the time increment $(t_2 - t_1)$ along its trajectory).

matches of air mass parcels (5 hPa vertical and 13-15 km horizontal size) at different altitudes have been obtained during these
two consecutive HALO flights.

The procedure described above was applied to all cases when two HALO flights took place on consecutive days. This included not only WAIs or atmospheric rivers (ARs), but also CAOs, Arctic cirrus (AC) and polar low (PL) cases (Walbröl et al., 2023). The respective statistics of the number of quasi-Lagrangian matches are given in Fig. 4a. Numerous cases have been identified where individual air mass parcels, sampled on day one have been encountered a second time on day two.
In a similar way, individual flights performed during a single day were analyzed (Fig. 4b). In these cases, matches between different flight sections along the individual flight were identified. Overall, the number of matches in both scenarios (two flights on consecutive days, one flight during one day) is more than sufficient for a statistical evaluation, scientific analysis, and discussion. This data set of quasi-Lagrangian matches provides a new quality of possibilities to observe and study atmospheric air mass transformations.

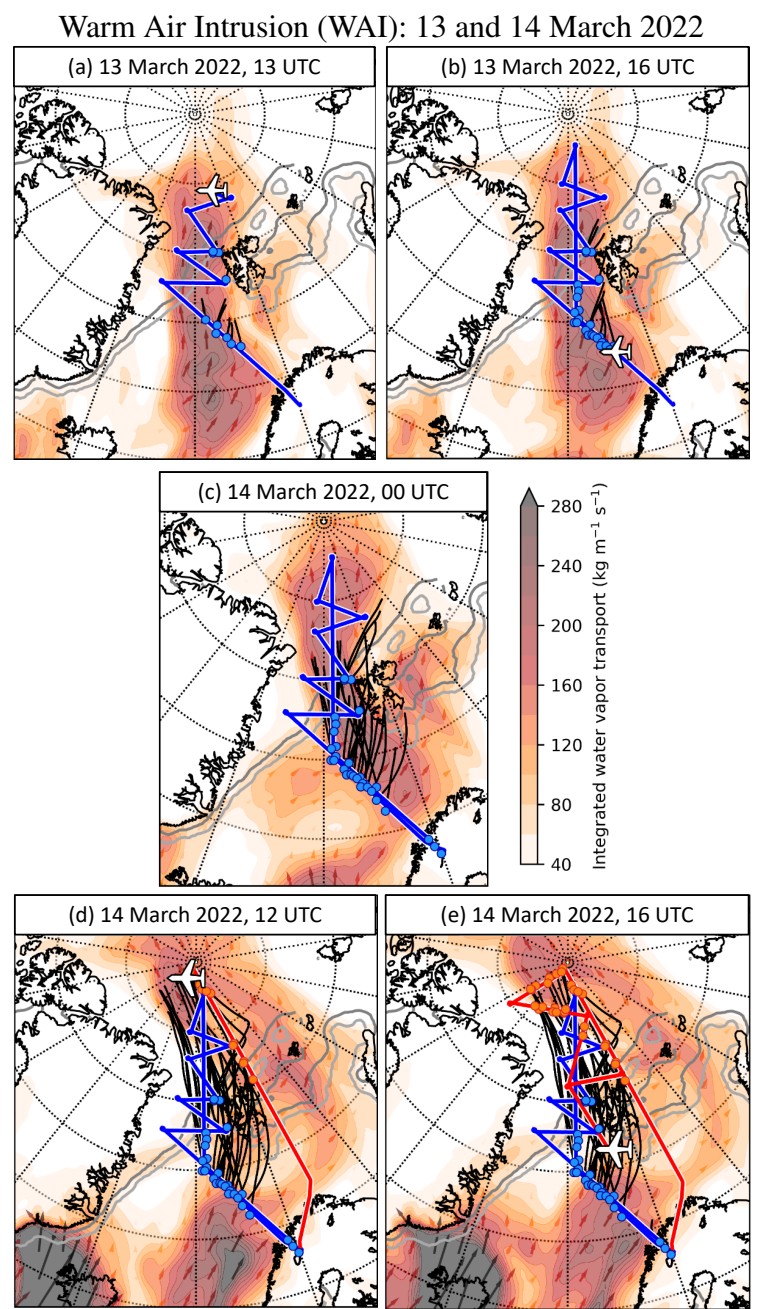

**Figure 3.** Time series (based on ERA5) of geographic maps including the integrated water vapor transport (IVT, in red colors, with the length of arrows being proportional to the magnitude of IVT), and the HALO flight tracks on 13 March 2022 (RF03, blue line) and 14 March 2022 (RF04, red line). Black lines indicate the horizontal projections of the evolving matching trajectories. The blue filled circles represent the trajectory start points, and the filled orange circles show where trajectories cross the second flight track.



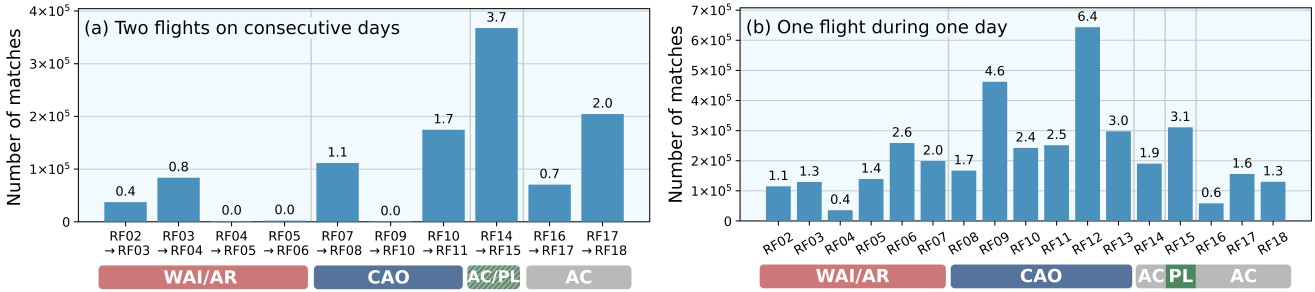

**Figure 4.** Number of matches for the two scenarios: (a) Two flights on consecutive days, and (b) one flight during one day. The analysis includes moist and warm air intrusions (WAIs), atmospheric rivers (ARs), marine cold air outbreaks (CAOs), as well as Arctic cirrus (AC), and polar low (PL) cases.

## 4 Initial results and ongoing analysis

In this section, we provide the first results obtained from the measurements conducted during the HALO–$(\mathcal{AC})^3$ aircraft campaign with respect to air mass transformations during CAOs and WAIs, Arctic cloud and aerosol characteristics, and atmospheric turbulence measurements. Furthermore, we discuss a proof of concept to measure mesoscale divergence and subsidence in the Arctic. In the three appendices, we add partly preliminary, but nonetheless very interesting supplementary discussion and results from the HALO–$(\mathcal{AC})^3$ campaign: Appendix A: A hypothesis on humidity fields in WAIs, Appendix B: The particular mode structure of the Arctic radiant energy budget, and Appendix C: An extraordinary CAO case covered during HALO–$(\mathcal{AC})^3$.

### 4.1 Air mass transformations during CAOs and WAIs

#### 4.1.1 Thermodynamic tendencies

To quantify the tendencies of thermodynamic properties of air mass parcels, such as the dry potential air temperature ($\theta$), specific humidity ($q$), and air temperature ($T$), we apply the quasi-Lagrangian match analysis using Eq. 1 with $\psi = \theta, q, z, T$. Corresponding results are depicted in Fig. 5. Here, the vertical profiles of the numbers of occurrence (counts) of certain values of the $\theta$, $q$, and $T$ tendencies are illustrated (Eq. 1). In addition, the vertical displacement of the air mass parcels along its trajectory between $t_1$ and $t_2$ is plotted. The tendencies for the CAOs (Figs. 5a–d) are derived for trajectories from the north to south, i.e., warming and moistening at low levels occurs due to heat and moisture surface fluxes increasing from sea ice to open ocean. The WAI tendencies (Figs. 5e–h) concern the opposite sense, i.e., from south to north. It is important to note that the diabatic heating and moistening presented in Fig. 5a-d merge the quasi-Lagrangian matches over open ocean exclusively (CAOs), whereby all available matches (over open ocean and sea ice) are considered in the tendencies for WAIs.

Instead of the air temperature $T$, we first investigate the change rate of the dry potential air temperature $\theta$, which is insensitive to dry adiabatic vertical movements of the air mass parcel during transport. Furthermore, $\theta$ is characteristic of a moving air



mass and changes only in response to diabatic processes. These include cloud evolution (release or consumption of latent heat), surface influences (such as turbulent and energy fluxes), as well as radiative processes. Using $\theta$ instead of $T$ thus quantifies the influence of processes we are most interested in, namely the cloud and surface effects.

For the CAO cases (panels a and b), within the layer between the surface and about 0.5 km altitude surface-driven diabatic
heating between 1–3 K h$^{-1}$ (Fig. 5a) and a moistening between 0.05–0.3 g kg$^{-1}$ h$^{-1}$ (Fig. 5b) are observed. Air mass parcel trajectories are descending throughout most of the vertical column with a wider spread of upward and downward motion in the atmospheric boundary layer (ABL) (Fig. 5c). For the WAI observations, a weak diabatic cooling of up to 0.4 K h$^{-1}$ (Fig. 5e), and a moisture loss of up to 0.1 g kg$^{-1}$ h$^{-1}$ (Fig. 5f) are observed, both reaching from the surface to heights to about 5 km. For a specific intense CAO (RF12; 1 April 2022, not shown here), Kirbus et al. (2023b) derived a maximum diabatic heating larger
than 6 K h$^{-1}$ close to the ocean surface just downwind of the MIZ. Values of moisture uptake of more than 0.3 g kg$^{-1}$ h$^{-1}$ were observed in this study from Kirbus et al. (2023b).

For the CAOs, slight subsidence of the air mass parcels during transport is derived (Fig. 5c). If air temperature tendencies are evaluated instead of dry potential air temperature $\theta$, the adiabatic warming effects due to subsidence for the CAO cases become apparent throughout the entire vertical column (Fig. 5d), whereas no such adiabatic warming is obvious in Fig. 5a. No
clear ascent/descent trends of the air parcels are obvious from Fig. 5g.

Similar to Fig. 5, the cloud reflectivity $Z$ measured by radar on HALO was used to follow the cloud evolution during CAOs and WAIs (not shown). In case of CAOs, clouds evolve mainly in lower altitudes (below 3 km). In case of WAI, cloud dissipation dominates mostly below 6 km altitude.

### 4.1.2 Development of cloud properties during CAOs

Firstly, we investigated how the preconditions over sea ice influence the crucial initial cloud formation using targeted research flights by the P5 which statistically sampled the developing roll convection just behind the MIZ (80–100 % sea ice concentration). Contrary to previous airborne studies that followed the air mass downstream along developing cloud streets, the P5 flew multiple legs orthogonal to them. In this way, the roll convection forming the cloud streets could be identified from radar profiles and characterized statistically with respect to their macrophysical and microphysical cloud properties using multiple
instruments (Schirmacher et al., 2024). Because air mass transformation is mainly triggered by the exposure of air to open water surfaces, we used backward trajectories to assign each measurement to its fetch, i.e., the horizontal distance the air mass traveled over open water until reaching P5. Two cases of CAOs with many similarities but different strengths observed on the 1st and 4th of April 2022 were analyzed in detail by Schirmacher et al. (2024) and are summarized briefly here. The evolution of crucial parameters characterizing the cloud and precipitation development along the two CAOs as a function of fetch is
illustrated in Fig. 6 within the first 170 km. Cloud streets form approximately 15 km downwind of the sea ice edge reaching almost 100 % cloud cover for a fetch exceeding 20 km in both cases. Due to the strong surface fluxes, the ABL grows quickly, and cloud top height (CTH) increases by about 4 m per km in the strong CAO case (1 April 2022) and only half as much for the weaker CAO (4 April 2022). Both events feature mixed-phase clouds with the stronger case showing twice the liquid water path (LWP). Precipitation sets in after a fetch of about 30 km with a slightly later onset for the weaker event. The combination





**Figure 5.** Vertical profiles of the number of occurrences (counts) of temporal tendencies of (a) dry potential air temperature (diabatic heating, $\Delta\theta/\Delta t$), (b) specific humidity (moistening, $\Delta q/\Delta t$), (c) air mass parcel ascent ($\Delta z/\Delta t$), and (d) air temperature ($\Delta T/\Delta t$) for marine cold air outbreaks (CAOs) sampled with HALO on 20, 21, 28, 29, 30 March and 1 April 2022. Panels (e) to (h) illustrate the same temporal tendencies for moist and warm air intrusions (WAIs) observed with HALO between 12-16 March 2022.

of remote sensing from P5 with in-situ measurements by P6 also allows us to investigate the ice growth process (Maherndl et al., 2024). For the stronger CAO case, we also detect stronger riming which occurs on a horizontal scale similar to the roll circulation.

    To understand how CAOs develop from their initial phase along their way further south, HALO with its long range can provide valuable insights. For the stronger CAO, images of spatial CTH and cloud thermodynamic phase were gathered the

spectrometer of the Munich Aerosol Cloud Scanner (specMACS) instrument (Ewald et al., 2016; Weber et al., 2024) west of

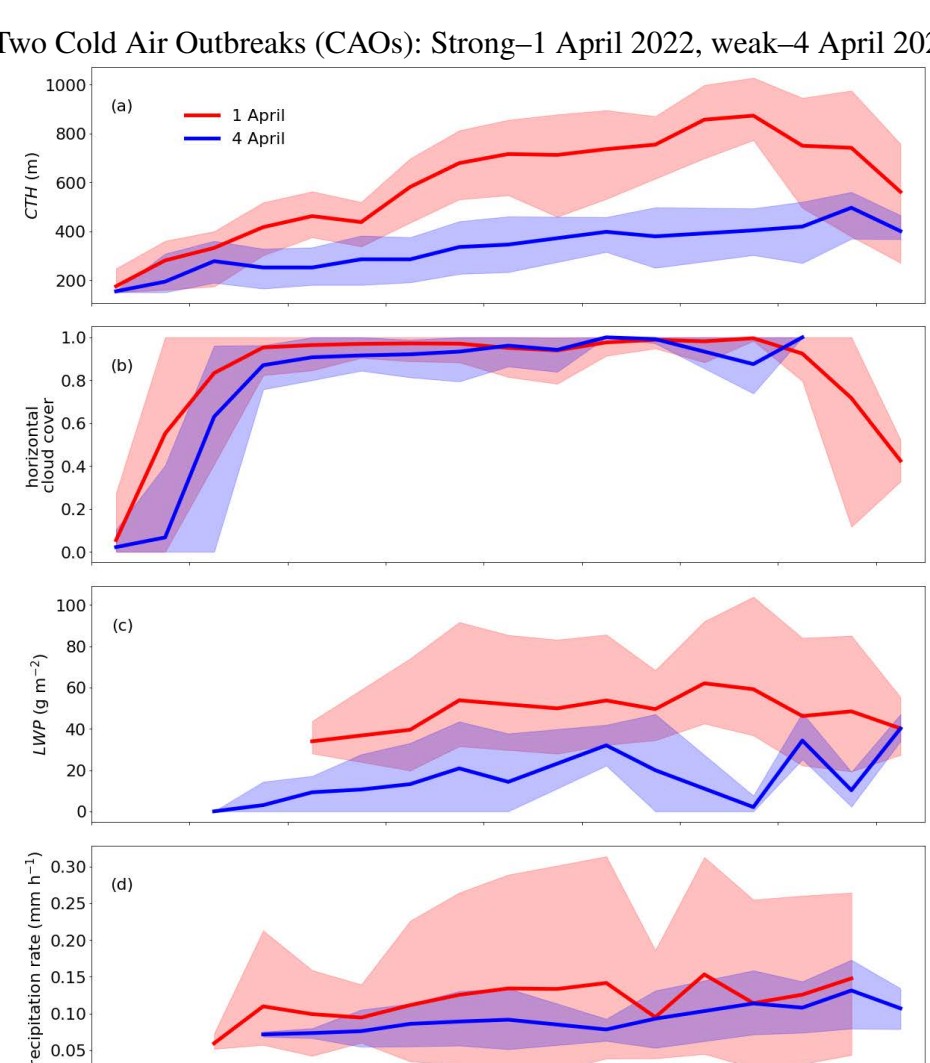

**Figure 6.** Development of geometrical and microphysical cloud properties as a function of fetch on 1 April (red line, strong CAO) and 4 April (blue line, weak CAO) 2022 as measured with P5: (a) Cloud top height (CTH), (b) horizontal cloud cover per minute measured by the radar MiRAC and lidar AMALi, (c) liquid water path (LWP), and (d) precipitation rate at 150 m height ($S$). The shaded areas indicate the 5 % and 95 % quantiles of the distributions with fetch. Figure adapted from Schirmacher et al. (2024).

Svalbard during HALO RF12. The spectral slope phase index defined by Ehrlich et al. (2008) was used to derive information about cloud thermodynamic phase from the measurements. Values of the spectral slope phase index smaller than about 20 indicate pure liquid water, and larger values mixed-phase or ice clouds. In addition to the thermodynamic phase, we have retrieved CTH from the specMACS observations with a stereographic method (Kölling et al., 2019; Volkmer et al., 2024). This

data was again combined with back trajectories to calculate the time the measured air mass traveled above the open ocean after



passing the MIZ (here 80 %). Figure 7 shows three example images of the polarization cameras showing how the clouds in their initial phase organize in cloud streets and then develop into closed cells with increasing distance to the MIZ. While the initial phase with cloud tops up to 1 km was already captured by P5, the HALO observations show that CTH continues to increase up to 2 km. A transition from pure liquid water to mixed-phase clouds occurs within the first two hours after passing the MIZ.

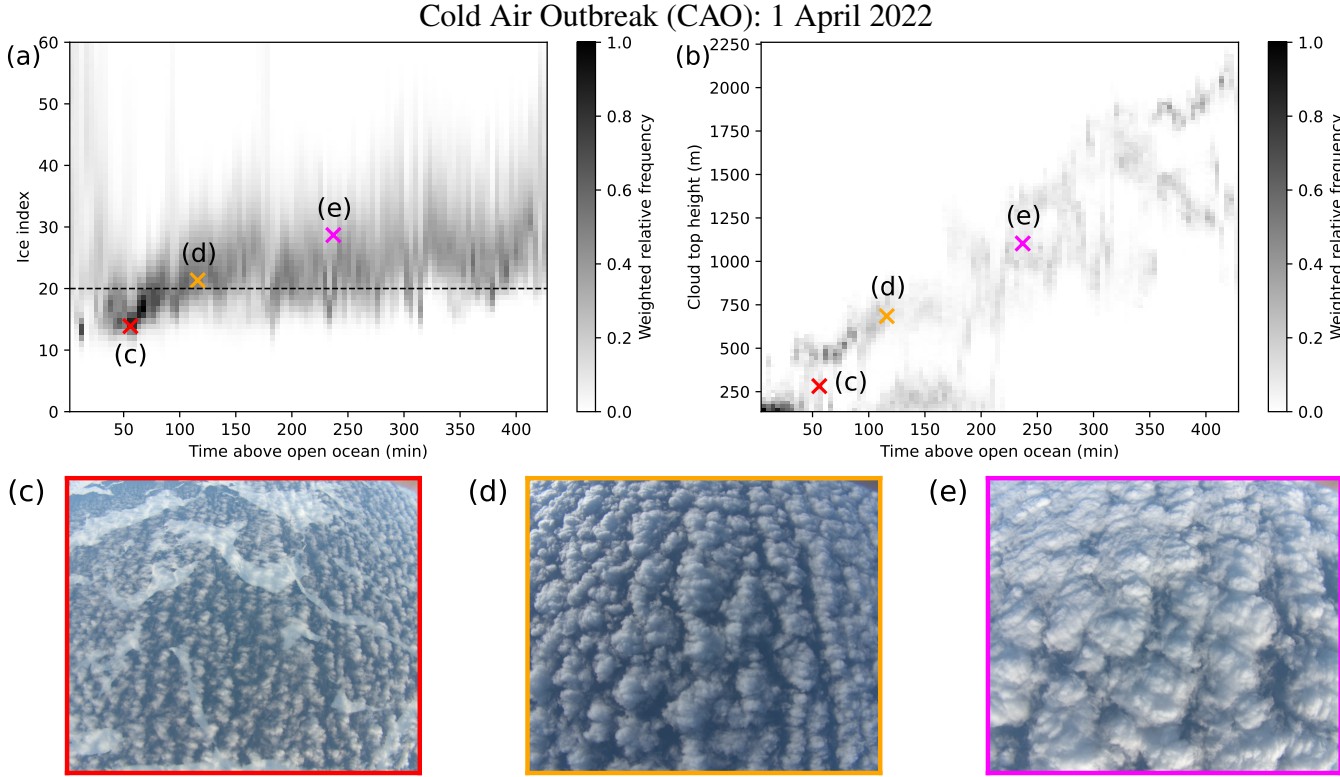

**Figure 7.** (a) Spectral slope phase index (ice index) derived from measurements of the spectrometer of the Munich Aerosol Cloud Scanner (specMACS) on 01 April 2022 as a function of the time the air mass traveled above open ocean. (b) Same as (a) but with cloud top height from stereographic reconstruction. (c) to (e) Example RGB (red-green-blue) images at points indicated in panels (a) and (b).

But how much liquid and ice water is contained in CAOs? von Lerber et al. (2022) noted the difficulty of quantifying snowfall in CAOs both from models, which heavily rely on parameterizations, and also in spaceborne radar, as CAO clouds are often within the blind zone of Cloudsat. LWP is a key parameter within the energy and water cycle. Therefore, it is important to understand how LWP and its spatial distribution develop during CAOs. However, observing LWP is prone to high uncertainties, especially in the Arctic, leading to about a factor of two difference in satellite retrievals between passive microwave and solar

radiation retrievals (Lohmann and Neubauer, 2018). Our LWP measurements from P5 and HALO offer opportunities to better constrain satellite observations. Exemplary, Fig. 8b illustrates the flight path of HALO on 21 March 2022 (RF08) when HALO probed a CAO in its initial state close to the MIZ flying parallel to the sea ice edge, perpendicular to developing cloud rolls repeating the pattern with increased distance from the MIZ and finally straight towards Kiruna. To illustrate the quality of



different LWP data sets from space and aircraft, we focus on a flight leg where HALO sampled the transition to cellular

convection (Fig. 8c). Cloud radar measurements show a clear rise in CTH for this leg and growing cells southwards (at later

times). LWP values retrieved from the HAMP instrument (Mech et al., 2014) have maximum values around $300\,\mathrm{g\,m^{-2}}$ within

cells, while close to the sea ice edge, maximum values hardly reach $100\,\mathrm{g\,m^{-2}}$. They clearly resolve the individual cells which is

not possible from spaceborne microwave radiometry due to their coarse resolution. Note, that the HAMP retrieval only includes

the cloud contribution, and thus no enhanced values occur in precipitation. This leg was coordinated with the British FAAM

aircraft, which focused on in-situ measurements of the cloud and sub-cloud layer along this coordinated track enabling future

joint analysis. Due to the time shift between the MODIS (Moderate Resolution Imaging Spectroradiometer) measurement and

the HALO flight track, convective cells are shifted between MODIS and HALO measurements. Nevertheless, it becomes clear

that MODIS generally shows higher LWP values for these warm cloud conditions. Our comprehensive measurements from

different platforms will be used to further investigate the reasons for this long-standing problem.

The combination of various remotely sensed measurements with back trajectory calculation for the targeted CAO flight by

P5, P6, and HALO bears strong potential to further investigate CAO cloud development and transitions. Future analysis will

include further retrieval development of liquid and ice clouds, e.g., from specMACS, HAMP, and detailed evaluation of satellite

data. Most importantly, the comprehensive measurements provide solid reference data to test high-resolution models that are

able to resolve the complex circulation involved in CAOs.

### 4.1.3   Moisture budget during WAIs

An ultimate test of our understanding of the atmospheric water cycle is to be able to close the water budget. On 15 March

2022, a research flight (RF05) was dedicated to optimally determining the moisture budget components (Eq. 2), including their

accuracy for a strong WAI event. These data should serve as a critical test for the respective simulations using the ICON

(Icosahedral Nonhydrostatic) model. The governing equation for the moisture budget is given by the local change of the

integrated water vapor (IVW):

$$\frac{\partial\,IWV}{\partial t} = E - P - \overrightarrow{\nabla} \cdot IVT + \epsilon, \tag{2}$$

with $t$ the time, $E$ the evaporation, $P$ the precipitation rate, $IVT$ the integrated water vapor transport, and $\epsilon$ the residual.

$\overrightarrow{\nabla} \cdot IVT$ describes the divergence of the $IVT$; this quantity is derived from the sum of the dynamic mass divergence ($DIV_{\mathrm{mass}}$)

and the integral of the horizontal moisture advection ($ADV$).

The flight pattern (Fig. 9a) chosen to assess the moisture budget includes two legs perpendicular to the flow (thick light green

lines, cross-flow), and one internal leg (thick blue line). The moisture flux across the two cross-flow flight legs is estimated

from the wind and humidity observations provided by dropsondes (large triangles). From the difference between exported and

imported moisture fluxes determined by the dropsonde observations along the cross-flow flight paths, we estimate the internal

divergence of moisture flux ($\overrightarrow{\nabla} \cdot IVT$) as one key component of the atmospheric moisture budget. Along the internal leg,

measurements of radar, microwave radiometer, and dropsondes were used to derive precipitation rate ($P$), evaporation ($E$),



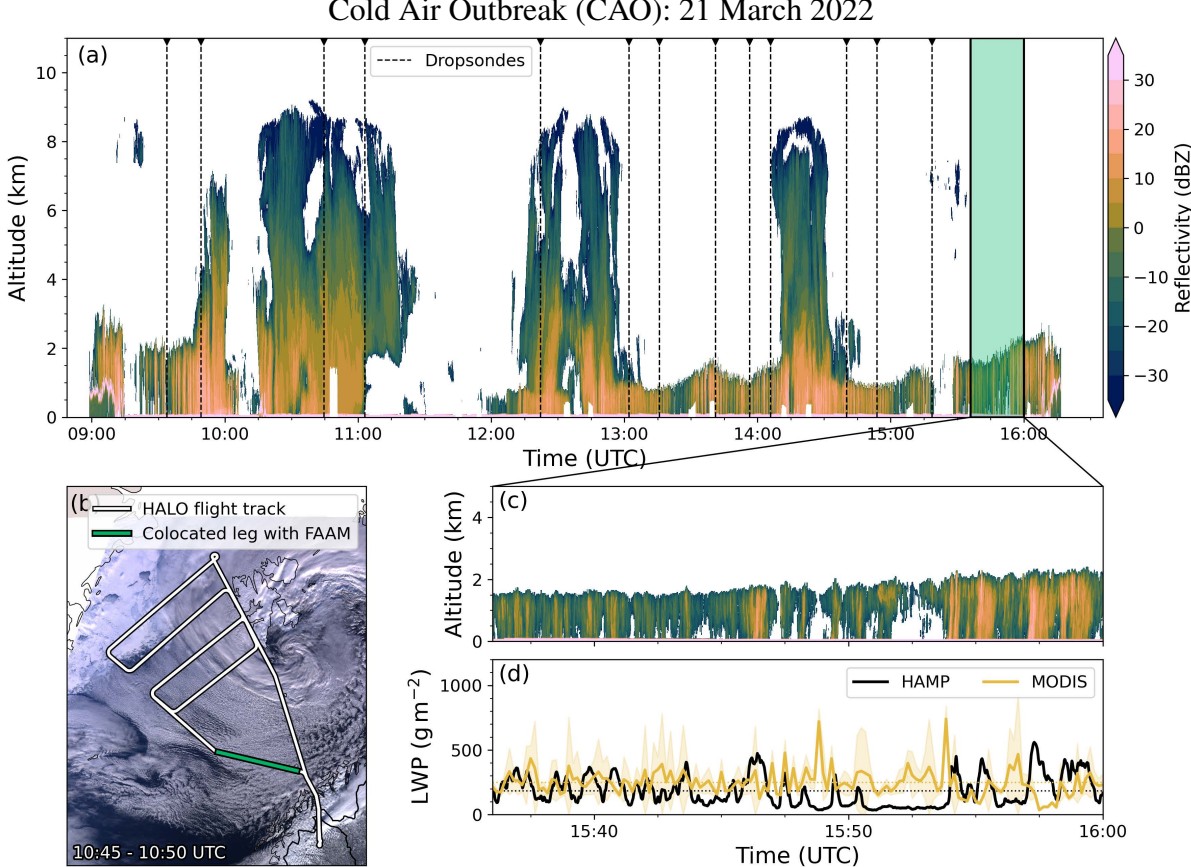

**Figure 8.** (a) Time series of radar reflectivity along the flight track of RF08 on 21 March 2022 with high reaching clouds belonging to the Shapiro-Keyser cyclone over Svalbard (Shapiro and Keyser, 1990), (b) Flight pattern of HALO and FAAM during RF08 on top of MODIS (Moderate Resolution Imaging Spectroradiometer) Terra image from 10:45-10:50 UTC (MOD02HKM - Level 1B Calibrated Radiances). (c) Zoom into radar measurements from last leg, which was collocated with FAAM (green line). (d) Time series of liquid water path (LWP) retrieved from HALO and MODIS (MOD06 - Cloud Optical Properties, two-channel retrieval using band 7 (2.1 $\mu$m) and band 6 (1.6 $\mu$m)) along the zoom of flight track of RF08. Dotted lines show the temporal mean.

integrated water vapor ($IWV$), and its temporal tendency (d $IWV$ / d $t$). The simulations with ICON are performed in the domain enclosed by the red dashed line in Fig. 9a with a horizontal resolution of 2.4 km.

Fig. 9b compares the moisture budget components derived from the HALO observations (full triangles) and ICON simulations (full dots) along the HALO track. The ICON-based and observational estimates derived agree reasonably well in the quantification of moisture tendency due to mass convergence and surface evaporation, while there are discrepancies regarding the temporal tendency of water vapor. A potential explanation might be the substantial dissipation of the WAI during the flight. Future work will focus on understanding the causes of these discrepancies, closing the moisture budget in the observations, and identifying the major processes for the correct representation of WAIs in models. Here we can also exploit RF02, RF03,




RF04, and RF06 where meteorological conditions and the flight pattern of HALO are well suited to estimate the local moisture
tendency and directly compare it with the ICON simulations.

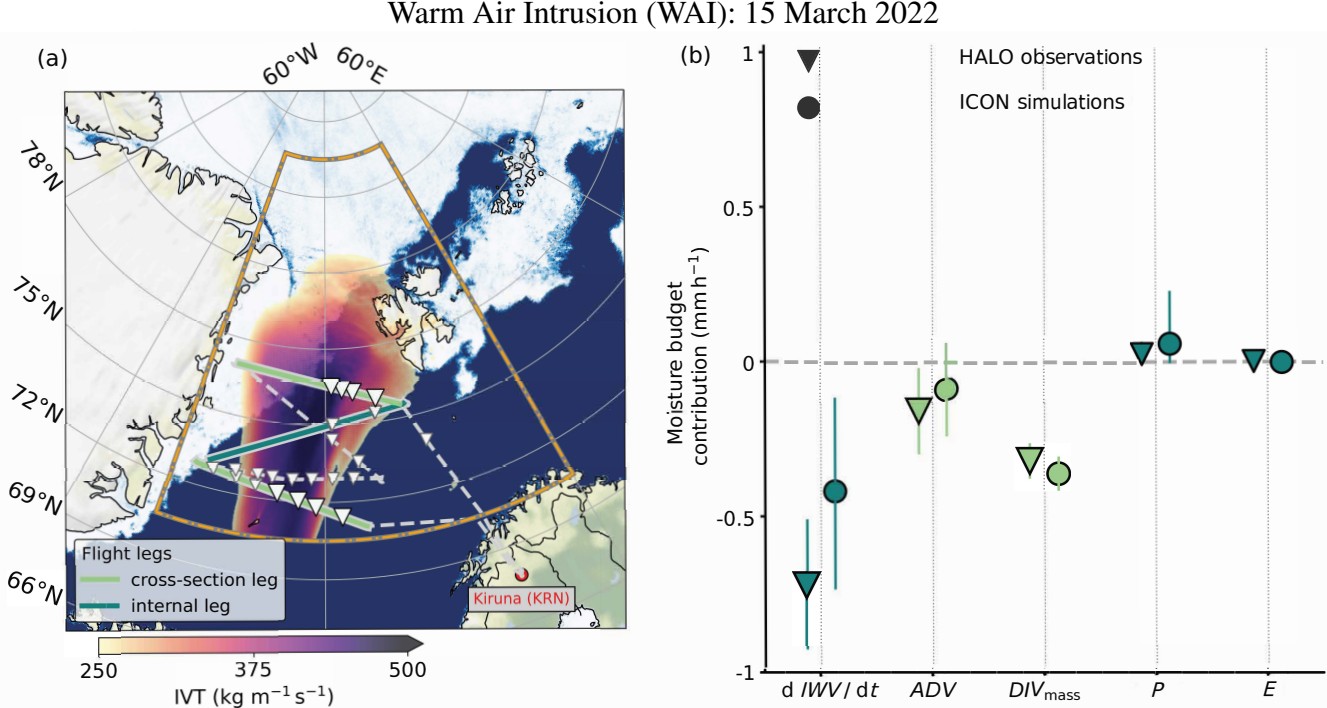

**Figure 9.** (a) HALO flight pattern (gray dashed lines and colored thick lines) and ICON (Icosahedral Nonhydrostatic) model domain (framed by the orange dashed line) for the WAI sampled during RF05 on 15 March 2022. Along the two cross-flow flight sections (thick light green lines), dropsonde observations are indicated by large triangles, whereas small triangles represent the dropsonde releases during the remaining flight sections. The internal leg is indicated by the blue solid line. For the ICON domain, simulated integrated water vapor transport ($IVT$) values are illustrated in red color. (b) For the eastern part of the flight pattern, the moisture budget components from the HALO observations (triangles) are compared to those simulated by ICON (circles), with respect to their contribution to the moisture budget in mm h$^{-1}$. Moisture advection ($ADV$), mass divergence ($DIV_{\mathrm{mass}}$), precipitation rate ($P$), evaporation ($E$), integrated water vapor ($IWV$), and its temporal tendency (d $IWV$ / d$t$) are compared. Vertical lines indicate the uncertainties for each component.

## 4.2 Arctic clouds

### 4.2.1 Arctic low-level clouds: Thermodynamic phase distribution

The thermodynamic phase of the clouds is determined from the microphysical cloud properties derived from the particle size distribution data in the size range from 2.8 $\mu$m to 6.4 mm (Moser et al., 2023). The measured fractions of ice, mixed-phase, and
liquid water clouds are shown as a function of altitude in Fig. 10, classified into cloud measurements over open ocean and over sea ice. Due to the decrease of the temperature with altitude, the fraction of ice clouds over the ocean increases with altitude



for altitudes larger than about 500 m. Over sea ice, the ABL shows a high fraction of ice and liquid water. In contrast, the cloud characteristics over the ocean are more variable in height as mixed-phase clouds and pure liquid clouds are detected over the whole altitude range.

These results are obtained from cloud measurements conducted in different meteorological conditions, including CAOs, convergence lines, and polar lows. For a more precise statistical analysis of the cloud properties collected over the ocean, a normalized cloud structure is beneficial. Near future studies will evaluate microphysical properties, including the total number concentration, cloud particle effective diameter, and the cloud water content, and relate the data to environmental conditions. The results of further statistical thermodynamic and microphysical analyses on low-level Arctic clouds measured during

HALO–$(\mathcal{AC})^3$ will be compared with a previous data set collected in a similar season and checked for representativeness.

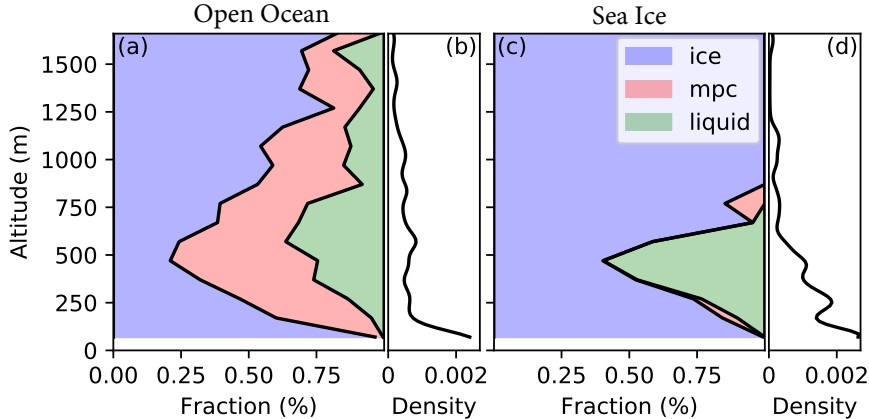

**Figure 10.** Fraction of detected cloud particle types resolved by altitude. Cloud types (ice: ice clouds; mpc: mixed-phase clouds; liquid: liquid clouds), shown for (a) clouds over the open ocean, and (c) clouds over the sea ice. (b) and (d) show the distribution of measurements in altitude. Thermodynamic phase classification was performed according to the algorithm presented by Moser et al. (2023).

To understand the conditions and feedback mechanisms, that maintain the persistence of the inherently unstable mixture of super-cooled liquid water cloud droplets and ice crystals, a three-dimensional characterization of the thermodynamic phase partitioning is required. For this purpose, the radar-lidar retrieval framework VarCloud was used to retrieve ice cloud microphysical properties of mixed-phase clouds (Aubry et al., 2024). An example for the simultaneous retrieval of cloud ice and

liquid water microphysical properties is given in Fig 11, which shows the cloud particle (liquid water droplets and ice crystals) effective radius retrieved from combined radar-lidar measurements collected during the HALO RF06 on 16 March 2022. The figure shows the cloud cross-section within a decaying WAI. On top of the leading marine stratocumulus deck and embedded in the trailing ice cloud layer, the combination of strong lidar with unremarkable cloud radar returns indicates the presence of layers of super-cooled water. These two regions with embedded super-cooled liquid water layers and liquid water-topped ice

clouds represent two distinct types of super-cooled liquid water. While the long-lived nature of the latter is well understood, the presence of super-cooled layers embedded within deep ice clouds requires further investigations that are planned for future work.



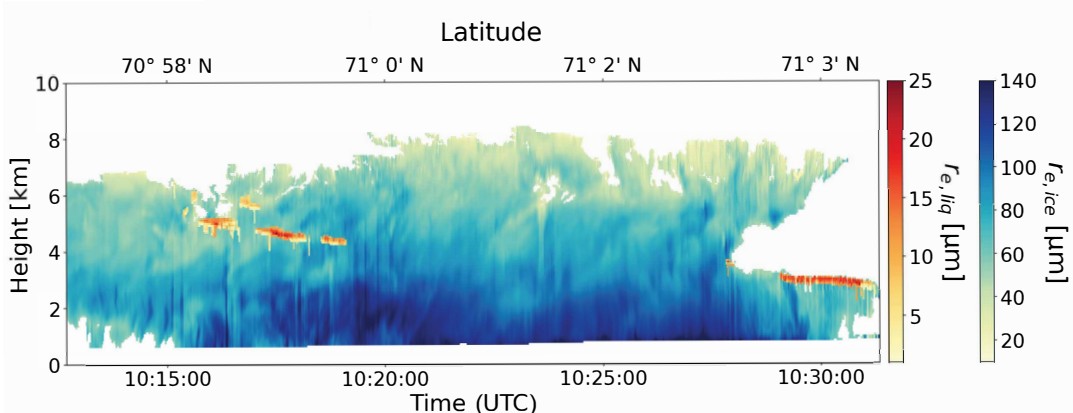

**Figure 11.** Simultaneous retrieval of effective radius of cloud ice crystals ($r_{e,ice}$) and liquid water droplets ($r_{e,liq}$) using radar-lidar measurements from the HALO research flight RF06 performed on 16 March 2022.

### 4.2.2 Arctic cirrus: The impact of surface, water vapor, and aerosol

The highly reflecting sea ice surface strongly modifies the radiative effects of clouds in both the solar and thermal-infrared spectral ranges (Stapf et al., 2020; Becker et al., 2023). Compared to the open ocean, the high surface albedo and low surface skin temperature of sea ice increase the relevance of surface properties for the cloud radiative effects. Figure 12a shows the brightness temperature field measured by the VELOX (Video airbornE Longwave Observations within siX channels) instrument (Schäfer et al., 2022) in a broadband wavelength channel ranging from $7.7\,\mu$m to $12.0\,\mu$m. For comparison, the time series of the broadband thermal-infrared net irradiance measured by the broadband (solar and thermal-infrared) irradiance sensor called the Broadband AirCrAft RaDiometer Instrumentation (BACARDI) (Ehrlich et al., 2024) is shown in Fig. 12b. The brightness temperature field shows a tendency to lower values during the first half of this flight section (up to $30\,$km distance), which is caused by an increased ice water path and the reduced emission by the cold cirrus. However, the structure of the sea ice is still imprinted in the measurements (e.g., at $10\,$km and $20\,$km distance) indicating the high transmissivity of the cirrus. The cirrus is thinning towards the end of the flight section. The warmer areas of open leads and the thin ice of nilas increase the emitted upward radiance beyond $40\,$km. The thermal-infrared net irradiance shows higher values over the cirrus due to a reduced emission compared to the warmer surface. This difference quantifies the top of the atmosphere warming effect of the cirrus, which reaches in this specific case up to about $30\,\mathrm{W\,m^{-2}}$. However, due to the hemispheric integrating view of BACARDI, the surface variability is not obvious in the broadband irradiance but still might impact the total cirrus radiative effect. Hence, to estimate the total cirrus radiative effect, cirrus and surface inhomogeneities need to be considered.

WAIs lead to an increase of relative humidity and enhance aerosol particle concentrations. Both components can impact the evolution of the cirrus radiative effect. WAIs also often contain enhanced number concentrations of aerosol particles. Both components can impact the evolution of cirrus radiative effects. Therefore, it is important to characterize cirrus clouds in the Arctic and their changes due to increased impact from mid-latitude air masses. Studies on the distribution of relative humidity





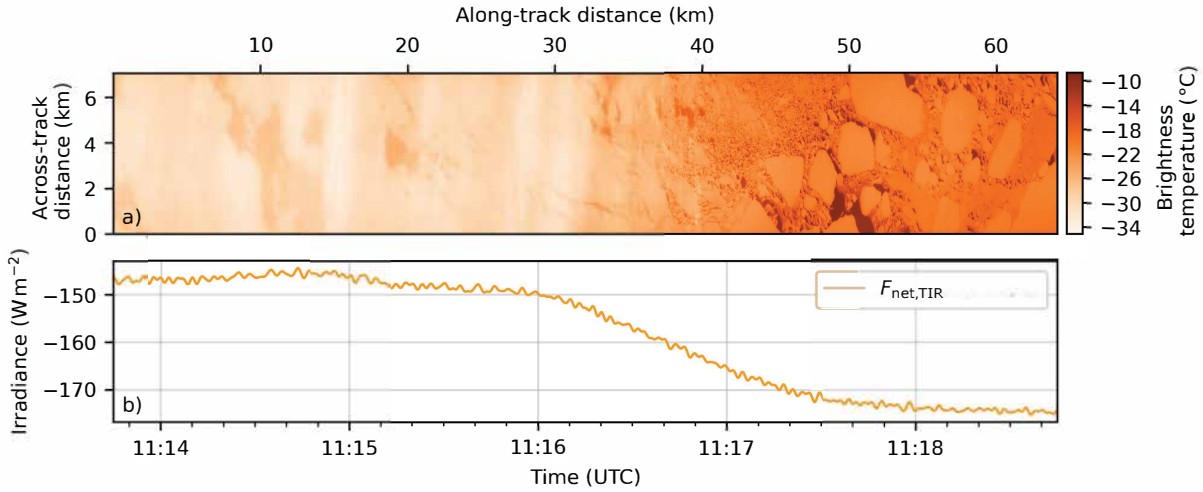

**Figure 12.** a) Two-dimensional brightness temperature field measured by VELOX along the flight track in the spectral range of $7.7\,\mu m$ to $12\,\mu m$. b) Time series of the thermal-infrared (TIR) net irradiance, $F_{\mathrm{net,TIR}} = F_{\mathrm{TIR}}^{\downarrow} - F_{\mathrm{TIR}}^{\uparrow}$, measured by the Broadband AirCrAft RaDiometer Instrumentation (BACARDI) for the same flight section.

with respect to ice (RHi) inside and around Arctic and WAIs cirrus have been performed using combined aerosol, cloud, and
water vapor measurements from the water vapor differential absorption lidar (WALES) (Wirth et al., 2009), together with
temperature information from the model analysis. Particular attention was paid to the vertical distribution of RHi within the
cirrus as well as on differences with respect to ice supersaturation giving an estimate of the dominant ice formation processes.
From the vertical profiles of cloud properties and RHi we found that cirrus formed in air masses transported into the Arctic by
WAIs have a larger vertical extent compared to cirrus formed in Arctic air masses (Dekoutsidis et al., 2023).

WAI cirrus are characterized by high ice supersaturation throughout their whole vertical profile. Fig. 13 shows an example of
the vertical extent and the RHi within and around a WAI cirrus measured during RF03 on 13 March 2022. From the backscatter
ratio (upper left) one can derive the vertical extent of the cirrus. A typical value to distinguish cloud from cloud-free pixels
is around three (Groß et al., 2014; Urbanek et al., 2017). Applying these values one can see that the WAI cirrus can have a
vertical extent from about 3 km altitude to about 12 km altitude. The cirrus is associated with large values of RHi (lower left).
Values of 140 % and larger are reached, and the majority of data points within the cloud shows supersaturation with respect to
ice. This becomes even more visible when looking at the combined distribution of backscatter ratio and RHi (Fig. 13, right).
High values of RHi within the cloud (backscatter ratio > 3) are found with a peak of the RHi distribution at about 110 %. Even
values exceeding the threshold of homogeneous freezing have been found inside and around the WAI cirrus.

## 4.3 Arctic aerosol particles

Sources, abundance, and properties of Arctic aerosol particles in general, and cloud condensation nuclei (CCN) in particular, are
not comprehensively monitored. Therefore, aerosol measurements were performed during the HALO–$(\mathcal{AC})^3$ aircraft campaign



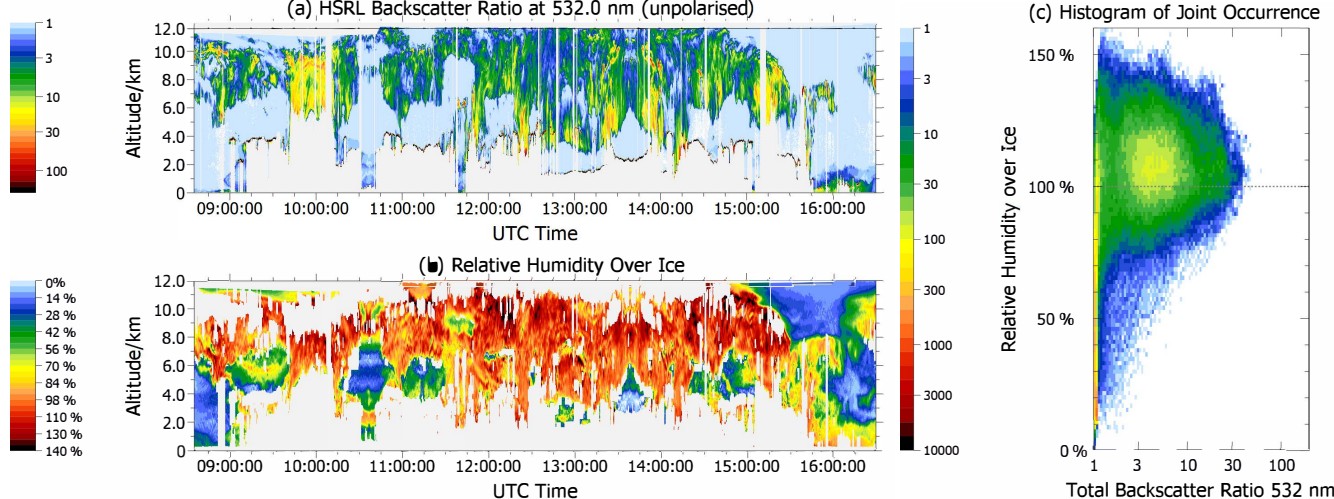

**Figure 13.** Cross-section of (a) backscatter coefficient at 532 nm, and (b) relative humidity with respect to ice (RHi) for RF03 on 13 March 2022 during a WAI. (c) shows a histogram of joint occurrence of RHi and backscatter coefficient at 532 nm. The relative humidity was calculated from WALES water vapor measurements and model temperature field.

using data obtained from in-situ aerosol instrumentation installed on board of P6. Particles were sampled behind a well-characterized aerosol inlet (Leaitch et al., 2016) and a Counterflow Virtual Impactor (CVI) (Ogren et al., 1985), showing comparable sampling characteristics when both were operated as an aerosol inlet (Ehrlich et al., 2019). Among others, in-situ

black carbon (BC) measurements were performed by a single particle soot photometer (SP2) installed behind the CVI. Further details on the applied instrumentation are given by Ehrlich et al. (2024).

We have derived typical values of microphysical aerosol properties measured during HALO–$(\mathcal{AC})^3$. We present averaged data from the research flights RF08–RF13 of P6 (1–10 April 2022), including periods over open ocean and sea ice. The analysis shows a median total aerosol particle number concentration $N_{\text{total}}$ of 303 cm$^{-3}$ (inter-quartile range (IQR): 207–419 cm$^{-3}$;

Figs. 14a and 14d), and a median CCN number concentration $N_{\text{CCN}}$ (measured at 0.1 % supersaturation) of 155 cm$^{-3}$ (IQR: 81–204 cm$^{-3}$; Figs. 14b and 14e). No obvious change of particle number concentration with altitude is evident up to roughly 1000 m above sea level, and only a slight decrease with height can be observed above. Average CCN hygroscopicity ($\kappa$ measured at 0.1 % supersaturation, Figs. 14c and 14f is in the range typical for mostly inorganic aerosol mixed with organic material (median: 0.50; IQR: 0.40–0.68), featuring slightly higher values above 1000 m.

Furthermore, we have measured cloud droplet residuals (CDR) from in-cloud sampling using the CVI installed on board of P6 (Ehrlich et al., 2019). We have compared the CDR aerosol properties collected in the ABL (below cloud) and in the free troposphere (above cloud). A size distribution example representative for the HALO–$(\mathcal{AC})^3$ conditions for low clouds over open ocean is shown in Fig. 15. Due to the identical shape of the CDR and particle distributions below cloud, it is concluded that the cloud is formed and sustained by the activation of ABL particles at the cloud base. A major influence of cloud-

forming particles entrained from the free troposphere above the cloud can be excluded since the respective size distribution



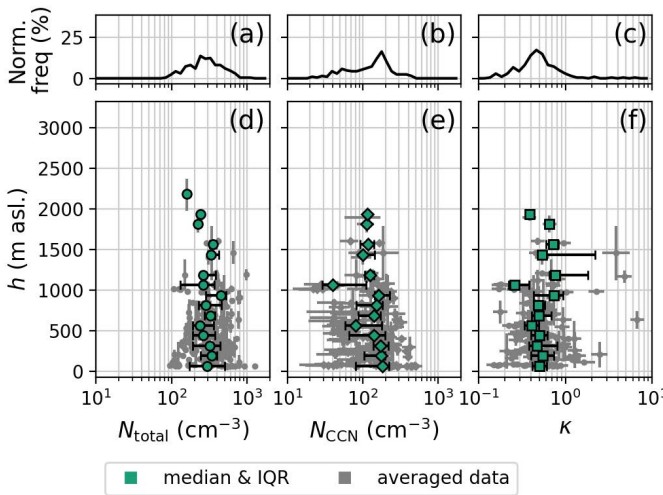

**Figure 14.** Relative occurrence of (a) total aerosol particle number concentration $N_{\text{total}}$, (b) Cloud condensation nuclei (CCN) number concentration $N_{\text{CCN}}$, and (c) CCN hygroscopicity $\kappa$. Vertical-binned averages of (d) $N_{\text{total}}$, (e) $N_{\text{CCN}}$, and (f) $\kappa$. P6 research flights RF08 to RF13 are considered.

appears different. This finding is similar to the ACLOUD results over open ocean (Wendisch et al., 2019). However, the absence of clouds over sea ice during HALO–$(\mathcal{AC})^3$ did not allow a comparison with ACLOUD results, where entrainment of cloud-forming particles from the free troposphere was suggested. This analysis will be continued to look for dependencies on the distance to the sea ice edge, the chemical composition of CDR and out of cloud particles, and changing meteorological
conditions during the HALO–$(\mathcal{AC})^3$ campaign.

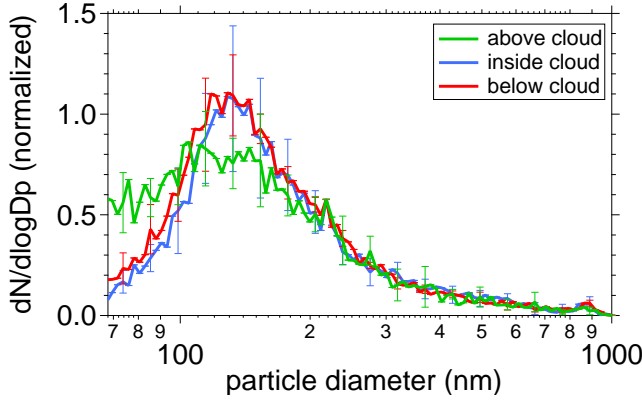

**Figure 15.** Normalized number size distribution of cloud droplet residuals (CDR) and ambient aerosol particles measured inside (blue), below (red) and above (green) clouds over open ocean during research flight RF01 with P6 (20 March 2022).



We also look at the chemical composition of aerosol particles and cloud residuals and their variation with season. For this purpose, we compare observations collected during ACLOUD (late spring/early summer) (Wendisch et al., 2019) with our data available from HALO–$(\mathcal{AC})^3$ (late winter/early spring). The single particle mass spectrometer ALABAMA (Brands et al., 2011; Clemen et al., 2020) was used to measure the chemical composition and size of single particles in the size range of 0.23–

$3\,\mu$m (50 % cutoff diameter with respect to detection efficiency). The ALABAMA was connected to the CVI inlet allowing for sampling of CDR and ambient aerosol particles, dependent on the counter-flow settings. Simultaneous measurements of trace gases such as CO, $CO_2$, and $O_3$ were used to identify different air mass origins, e.g. to distinguish polluted from non-polluted air masses. The results (Fig. 16) show a large abundance of particulate amines in ambient air and a dominance of amines in CDR during late spring/early summer (ACLOUD), emphasizing the importance of marine biogenic sources for summertime Arctic

cloud processes. In contrast, amine-containing particles were rarely observed during late winter/early spring (HALO–$(\mathcal{AC})^3$).

Furthermore, the abundance of elemental carbon (EC) containing particles was higher during HALO–$(\mathcal{AC})^3$, accompanied by higher CO mixing ratios than during ACLOUD, indicating the presence of anthropogenic pollution typical for the 'Arctic Haze' season in spring. These differences between the data obtained during ACLOUD and HALO–$(\mathcal{AC})^3$ reflect the general observation that ACLOUD was dominated by WAIs, while CAOs predominated during the HALO–$(\mathcal{AC})^3$ measurement flights

of P6. The composition of CDR is similar to that of the particles in ambient air during HALO–$(\mathcal{AC})^3$, but with a higher contribution of fresh and aged sea salt particles to the CDR, while in summer the amine-containing particles dominate the CDR. In future work, a detailed analysis of different air mass situations (WAIs versus CAOs) combined with air mass history analysis (e.g., air mass trajectories) will be included to investigate the sources of the identified particle types.

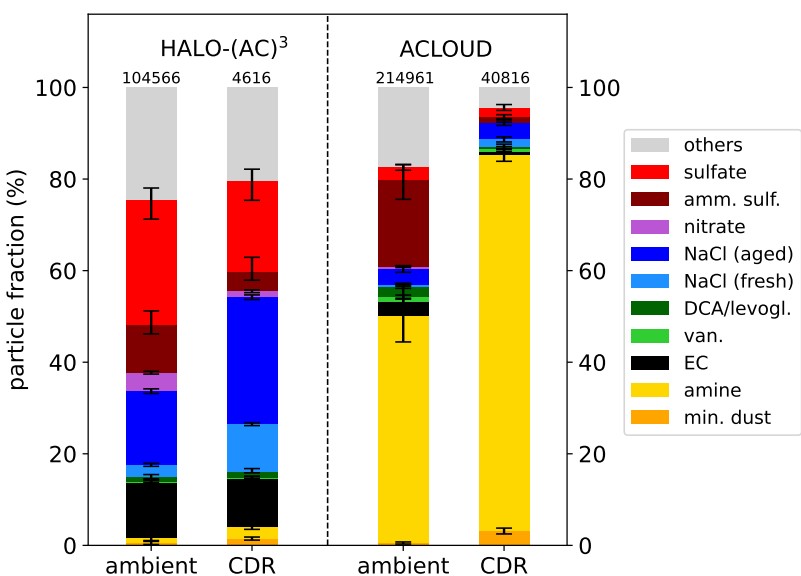

**Figure 16.** Number fraction of different particle types for ambient particles and cloud droplet residuals (CDR) during the aircraft missions ACLOUD and HALO–$(\mathcal{AC})^3$ analyzed by the ALABAMA. amm.sulf.: ammonium sulfate, DCA: dicarboxylic acids, van.: Vanadium, EC: elemental carbon. Particle types are assigned by the dominating peaks in the mass spectra.





### 4.4 Proof of concept to measure mesoscale divergence and subsidence in the Arctic

ABL cloud transformations at high latitudes play a key role in the Arctic and are partially controlled by large-scale dynamics such as divergence and subsidence. During several research flights (RFs) of HALO–$(\mathcal{AC})^3$, we have successfully applied a measurement technique using the data from multiple dropsonde releases in circular flight patterns to estimate mesoscale properties including divergence, associated subsidence, pressure gradients, and advective tendencies (Paulus et al., 2024). As illustrated in Fig. 17a, HALO RF10 and RF11 probed a weak CAO in a quasi-Lagrangian way, sampling it at multiple points

during two days along its southbound trajectory from the central Arctic into the Fram Strait. The four selected locations were determined before take-off, using trajectory estimates based on forecast data. Mesoscale flight patterns surrounding these locations were then incorporated in the HALO flight plan, in principle allowing the calculation of mesoscale gradients across the area (Bony and Stevens, 2019).

The scientific objective of this effort is twofold. Firstly, we aim to test the hypothesis that this novel sampling technique (the

sondes) can reliably yield mesoscale profiles of (thermo-)dynamic circulation properties also at high latitudes, in particular in cold, transforming low level air masses. Various studies have reported encouraging results with this method in marine subtropical areas (George et al., 2021; Bony and Stevens, 2019) and the mid-latitudes (Li et al., 2022). We show that the method works equally well at high latitudes, given the absence of large-scale weak temperature gradients (Charney, 1963; Sobel et al., 2001) and the (partially) associated high transience in synoptic weather. The second objective is to achieve a

dataset of circulation properties sampled by HALO at mesoscales that is suitable for driving scientific process models of clouds in CAOs along the trajectory. While CAO cases for Large-Eddy Simulations (LES) and Single-Column Models (SCM) have been generated before (de Roode et al., 2019), typically the observational data to realistically constrain the simulation in the upstream areas was lacking. Previous LES studies in the Arctic have shown a strong dependence of Arctic mixed layers on larger-scale forcings, in particular subsidence (Neggers et al., 2019; Dimitrelos et al., 2023), prioritizing the need

for such observational data. The HALO–$(\mathcal{AC})^3$ campaign supplies this crucial data for the first time, a unique opportunity for exclusively forcing scientific process model experiments for this weather regime with mesoscale data sampled along the trajectory. This has not been achieved before and would represent a step forward in anchoring Arctic process model studies in reality.

Figures 17b-e show the vertical profiles of subsidence as derived from dropsonde data from four "mesoscale circle" flight

segments that were flown at selected locations along the two-day trajectory. The northernmost circle was sampled by HALO RF10, while the southern three circles were flown a day later by RF11. The third HALO circle on the trajectory (RF11 C02) located just south of the sea ice edge was also probed in collocation by the P5 and P6 aircraft, providing additional in-situ measurements of clouds, turbulence, radiation, and aerosol properties. We obtain robust profiles of mesoscale subsidence at all four circle sites. For reference the measured profiles are cross-compared with ERA5 reanalysis data, reporting less-than-

optimal agreement in general. At circle RF11 C02 the subsidence profile sampled by HALO is reproduced by independent dropsonde data from the P5 aircraft, conducted an hour later in the same area and covering only the lowest few kilometers



below the P5 aircraft. This agreement, in particular, suggests that the HALO profiles of subsidence are realistic, and provides proof of principle for the applicability of the mesoscale dropsonde technique also at high latitudes.

The encouraging results obtained for this CAO case are fully described by Paulus et al. (2024). They motivate taking the next step and configure LES and SCM experiments for this case that are purely based on HALO measurements. Apart from divergence and associated subsidence, the data also yields profiles of horizontal pressure gradients (or geostrophic wind) and advective tendencies of temperature, humidity, and momentum. The in-situ and independent P5 and P6 data can well be used to evaluate the process model simulations, yielding a complete package for realistic model studies of this CAO. This research is currently in progress.

Cold Air Outbreak (CAO): 29 and 30 March 2022

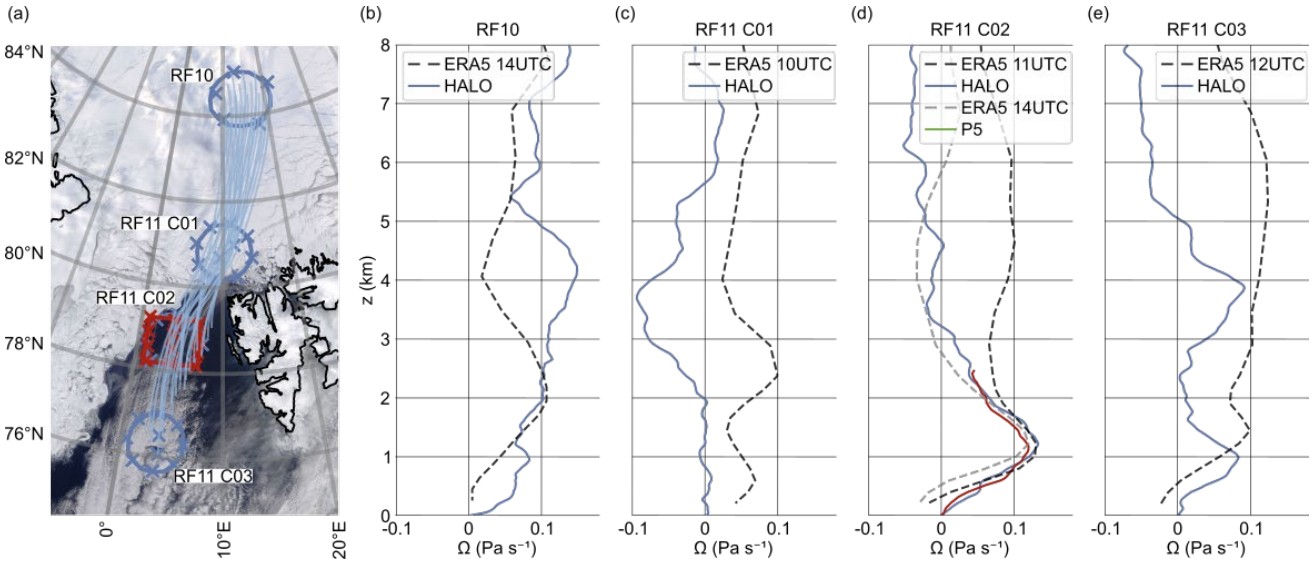

**Figure 17.** Overview of mesoscale flight patterns flown by HALO (dark blue) and P5 (green) sampling a low level air mass on 29 and 30 March 2022 at four locations along its southbound trajectory (light blue). Dropsonde launch locations are marked by crosses. b)-e) Profiles of pressure velocity $\Omega$ calculated from dropsonde data (colored) and ERA5 reanalysis data (grey dashed).

## 5   Summary

In this paper, we illustrate the application of a quasi-Lagrangian aircraft measurement approach to observe air mass transformations in the Arctic. The data were collected during the HALO–$(\mathcal{AC})^3$ aircraft campaign that took place between Scandinavia



and the North Pole in March and April 2022. We have employed three research aircraft with distinct objectives: The low-flying (mainly within and below clouds) Polar 6 (P6) was equipped with in-situ instrumentation to measure aerosol, cloud, and precip-
itation properties, whereas the higher-flying (mostly above clouds) Polar 5 (P5) aircraft conducted remote sensing observations of water vapor, aerosol particle, cloud, and precipitation characteristics. The third research aircraft, the High Altitude and Long Range Research Aircraft (HALO) was flying in about 10 km altitude. Whenever possible, HALO was flying closely collocated with P5 and P6. The in-situ and remote sensing payloads of the three aircraft were complemented by numerous dropsonde launches from P5 and HALO.

The focus of the observations was on air mass transformations during moist and warm air intrusions (WAIs) and marine cold air outbreaks (CAOs). The processes during these transformations were observed in a quasi-Lagrangian manner, probing the same air mass parcel twice on its way into or out of the Arctic, either on the same flight (e.g., at its beginning and end) or on two flights during consecutive days. To plan such flights, dedicated air mass parcel trajectory simulations along the flight paths were employed. A substantial number of matches of the same air mass parcels was obtained enabling statistical analysis of
the temporal tendencies of potential temperature and moisture during WAIs and CAOs. For the CAO cases, a strong surface-driven diabatic heating between $1$–$3 \, \mathrm{K \, h^{-1}}$ and a near-surface moistening between $0.05$–$0.3 \, \mathrm{g \, kg^{-1} \, h^{-1}}$ were observed within the lowest about 0.5 km. For the WAI observations, a weak diabatic cooling of up to $0.4 \, \mathrm{K \, h^{-1}}$ and a moisture loss of up to $0.1 \, \mathrm{g \, kg^{-1} \, h^{-1}}$ were obtained from the ground to about 5 km altitude.

    Furthermore, we followed the evolution of cloud properties (cloud top height and cover, liquid water path, precipitation rate,
thermodynamic phase, and radar reflectivity) during CAOs. We have shown that for a strong CAO case, cloud tops are higher, and more liquid water-topped clouds exist. The liquid water path and mean radar reflectivity increase compared to a weaker CAO. In addition, we see how the cloud parameters evolved with distance over the open sea, with the ABL deepening and CTH rising. We observed that for the stronger CAO case, the characteristic features such as the formation of cloud streets and the onset of precipitation occur closer to the sea ice edge.

Furthermore, the moisture budget of a WAIs was quantified for a strong WAI. ICON-based and observational estimates of the moisture budget components agree reasonably well for the moisture tendency due to mass convergence and surface evaporation, while there are remarkable discrepancies regarding the local tendency of water vapor in ICON compared to the HALO data.

    The vertical distribution of the thermodynamic phase in low-level Arctic clouds was quantified in a statistical manner using
in-situ cloud measurements carried out over the open ocean and sea ice. The clouds over sea ice are dominated by the ice phase. In the upper part of the ABL, liquid clouds are frequently detected. Over sea ice, only a small fraction of the observed clouds is attributed to mixed-phase clouds. In contrast, over the open ocean, the cloud phase distribution is more variable in height, as mixed-phase clouds and pure liquid clouds are detected over the entire altitude range. Some reasons for the longevity of mixed-phase clouds were discussed by using a three-dimensional characterization of the thermodynamic phase partitioning.

Typical aerosol characteristics were quantified, and the chemical composition of the aerosol particles was studied. Values of the median total aerosol particle number concentration of $303 \, \mathrm{cm^{-3}}$, and a median CCN number concentration at $0.1 \, \%$ supersaturation of $155 \, \mathrm{cm^{-3}}$ were derived. No altitude trend in particle numbers is evident up to roughly 1000 m above sea



level, but a decreasing trend was sometimes observed higher up. With regard to the chemical composition of the aerosol particles, a large abundance of particulate amines in ambient air and a dominance of amines in cloud droplet residuals (CDR)

during late spring/early summer is shown, emphasizing the importance of marine biogenic sources for summertime Arctic cloud processes. In contrast, amine-containing particles were rarely observed during late winter/early spring.

It was shown that circular flight patterns with sufficiently frequent dropsonde releases provide appropriate data to estimate mesoscale gradients, which can be used to derive subsidence and advective tendencies in the Arctic. These data are highly valuable, for example, to make available data to constrain the initial conditions for large eddy simulations avoiding the use of

numerical models with a coarser resolution.

Delivering data for evaluating and testing numerical models and reanalysis is a major objective of the HALO–$(\mathcal{AC})^3$ campaign. This objective will further be pursued using the wealth of observations from the campaign.

Analysis of the results of the HALO–$(\mathcal{AC})^3$ aircraft campaign is ongoing, and resulting papers will be published in a Special Issue of Atmospheric Chemistry and Physics (ACP) on "HALO–$(\mathcal{AC})^3$ – an airborne campaign to study air mass

transformations during warm-air intrusions and cold-air outbreaks", see: https://acp.copernicus.org/articles/special_issue1272. html.

*Data availability.* All data collected during the HALO–$(\mathcal{AC})^3$ aircraft campaign are being published by Ehrlich et al. (2024).

*Author contributions.* All authors provided text and figures and contributed to the editing of the article, and to the analysis and discussion of the results.

*Competing interests.* One of the co-authors (Birgit Wehner) is a member of the editorial board of ACP.

*Acknowledgements.* We gratefully acknowledge the funding by the Deutsche Forschungsgemeinschaft (DFG, German Research Foundation) – Project Number 268020496 – TRR 172, within the framework of the Transregional Collaborative Research Center "ArctiC Amplification: Climate Relevant Atmospheric and SurfaCe Processes, and Feedback Mechanisms $(\mathcal{AC})^3$". The authors are grateful to AWI for providing and operating the two Polar 5 and Polar 6 aircraft. We thank the crews and the technicians of the three research aircraft for excellent technical and

logistical support. The generous funding of the flight hours for the Polar 5 and Polar 6 aircraft by AWI, and for HALO by DFG, Max-Planck-Institut für Meteorologie (MPIM), and Deutsches Zentrum für Luft- und Raumfahrt (DLR) is greatly appreciated. We are further grateful for funding of project grant number 316646266 by DFG within the framework of Priority Program SPP 1294 to promote research with HALO. Oliver Eppers, Philipp Joppe, and Johanna Mayer acknowledge funding by the DFG – TRR 301 – Project-ID 428312742. Hans-Christian Clemen acknowledges funding by the DFG, project No. 442647984 (SCHN 1138/8-1). Timo Vihma acknowledges funding from the Euro-

pean Commission Horizon 2020 project Polar Regions in the Earth System project (PolarRES, 101003590). Gunilla Svensson and Michail



Karalis acknowledge funding from the Swedish Research Council (VR, project 2020-04064). Manuel Moser is grateful for funding by DFG SPP 2115 PROM (VO 1504/5-1), Johannes Lucke appreciates funding within the EU project SENS4ICE (grant agreement number 824253), and Elena De La Torre Castro acknowledges support by DFG SPP HALO 1294 (VO 1504/7-1). Harald Sodemann acknowledges funding from the European Research Council (Consolidator Grant Nr. 773245). This publication was supported by the Open Access Publishing Fund
of Leipzig University.



## Appendix A: A hypothesis on humidity fields in WAIs

We investigate the typical structure of the atmospheric moisture field, how it typically evolves during a WAI, and what processes drive this development. Relative humidity varies strongly and determines the structure of the moisture field in the cross-flow perspective going from 100 % in the core of the WAI to around 50 % within no more than 100 km distance form the core.

Temperature determines the structure of the moisture field in an along-flow direction, as relative humidity is at saturation throughout the core of the intrusion. In the ABL and close to the sea ice edge, temperatures change by 3-4 K over a distance of about 150 km. Clearly an atmospheric river is channeled in the Fram Strait between sea ice (20 °W) and Svalbard (0 °). In future work we will investigate the hypothesis that a secondary circulation acts to spread out moisture from the initial river-like intrusion in the cross-flow direction (Fig. 18). This hypothesized circulation consists of uplift in the core of the intrusions,

divergence in the upper troposphere above the core, and convergence and subsidence of drier air on the flanks of the intrusion.

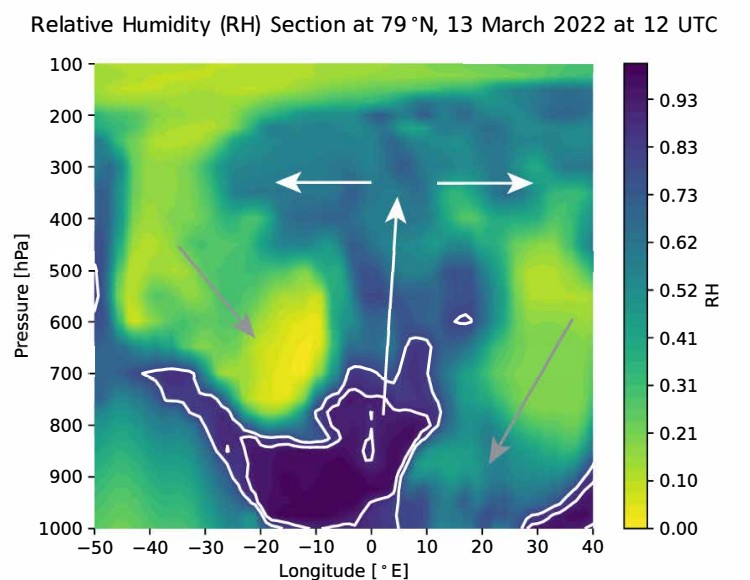

**Figure 18.** Zonal vertical cross-section of relative humidity with respect to liquid water during a moist and warm air intrusion (WAI) on the basis of ERA5 data. Arrows indicate hypothesized circulation.





**Appendix B: The particular mode structure of the Arctic radiant energy budget**

The radiant energy budget (REB) in the Arctic depends on different surface, cloud, and thermodynamic properties (Sedlar et al., 2011; Stapf et al., 2021). At the surface, Wendisch et al. (2023b) identified four modes (regimes) of the REB, depending on the surface type (sea ice or open ocean) and the atmospheric state (cloudy or cloud-free conditions). Models often struggle
to correctly represent this mode structure due to limitations in the treatment of sub-grid processes including clouds and the sea ice albedo (Kretzschmar et al., 2020; Solomon et al., 2023). Consequently, detailed observation–model comparisons are helpful to identify potential misrepresentations of properties affecting the REB.

During HALO–$(\mathcal{AC})^3$, the coordinated combination of measurements from three aircraft operating in different altitude regimes allowed to extend the statistical analysis of the Arctic REB to higher altitudes. Beside REB measurements performed
by P6 close to the surface, the REB above the ABL clouds and close to the tropopause was observed by P5 and HALO, respectively (Fig. 19). The common mode structure (cloudy and cloud-free, over sea ice and open ocean) is most prominently illustrated in the near-surface REB (modes indicated by 3, 4, and 5, Fig. 19c). Modes 4 and 5 result from the observations in cloudy and cloud-free conditions above P6 when flying over the open ocean. Mode 3 corresponds to cloud-free situations over sea ice (only few clouds were observed over sea ice). For P5 (mid-level regime, Fig. 19b), clouds were not present above the
aircraft, leaving a combined mode for weaker emitting cases above sea ice or low clouds (mode 1) and a mode for cloud-free observations above the open ocean (mode 2). The REB in high altitudes shows a continuous transition between modes 1 and 2. In general, the thermal-infrared net irradiance becomes more negative with increasing altitude. Furthermore, the surface and top of atmosphere (TOA) REBs obtained from ERA5 along the flight tracks are compared to the P6 and HALO measurements (not shown). The near-surface thermal-infrared REB is less negative than in the P6 observations. The TOA REB in ERA5 is
slightly shifted towards lower solar net irradiances and is more narrow than the distribution of the HALO observations due to the lower horizontal ERA5 resolution. Such comparisons will help to constrain the REB in numerical models which aim to reproduce HALO–$(\mathcal{AC})^3$ observations.



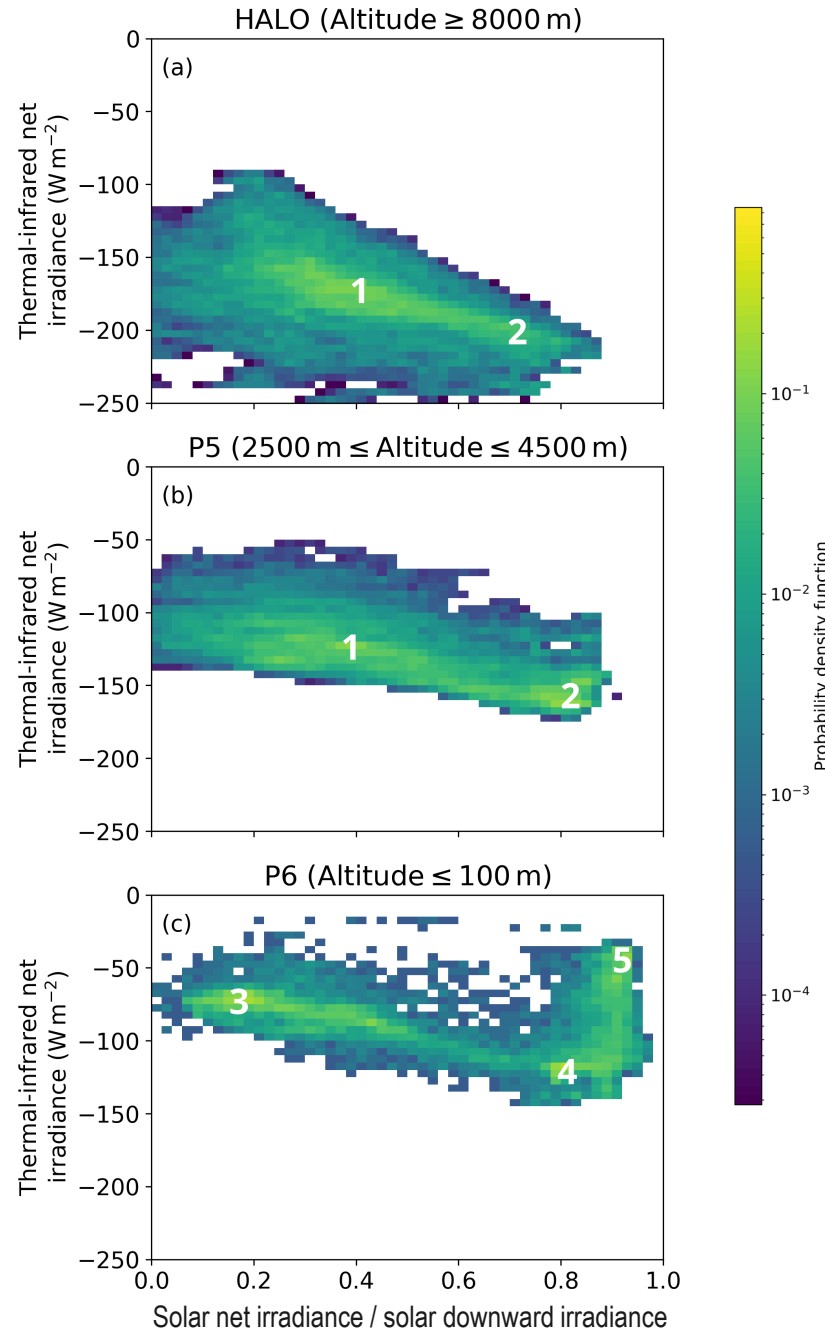

**Figure 19.** Two-dimensional probability density function of the solar (normalized by the solar downward irradiance) and thermal-infrared net irradiances observed in different altitudes: (a) HALO, (b) P5, and (c) P6 flight altitudes. Mode numbers: 1 – above clouds or sea ice, 2 – above open ocean, 3 – cloud-free over sea ice, 4 – cloud-free over open ocean, 5 – cloudy over open ocean.





## Appendix C: An extraordinary CAO case covered during HALO–$(\mathcal{AC})^3$

HALO–$(\mathcal{AC})^3$ has provided an opportunity to expand the limited data base of airborne CAO studies that are known from the
literature (Brümmer, 1997, 1999; Chechin et al., 2013; Dahlke et al., 2022; Geerts et al., 2022; Schirmacher et al., 2024). Here
we integrate our data obtained during two CAO events with measurements following approximately a North-South trajectory
(29 March and 9 April 2022) with literature data. The 9 April case was extraordinary for two reasons. Firstly, it was charac-
terized by cloud-free conditions along the entire 180 km North-South flight track across the MIZ to the open ocean, which is
very rare and unusual. Only at the southernmost position, convective clouds appeared. There and over sea ice, vertical profile
measurements were performed with P6 (RF12). Secondly, the southernmost position was influenced by the front of a polar
low, which was a remnant of a polar low over Fram Strait on the preceding day. Figure 20 shows profiles of mean and turbulent
thermodynamic quantities measured over open sea during the two HALO–$(\mathcal{AC})^3$ flights in comparison to previous measure-
ments. For 29 March 2022, the results fit well to the earlier measurements except some peaks in the momentum fluxes, which
could be traced back to orographic influences of Svalbard mountains at the northern coast. However, on 9 April (the cloudless
day), unusually high temperatures and untypically low wind speeds were observed leading to extraordinarily weak turbulent
convection over open sea when compared to previous CAO events.

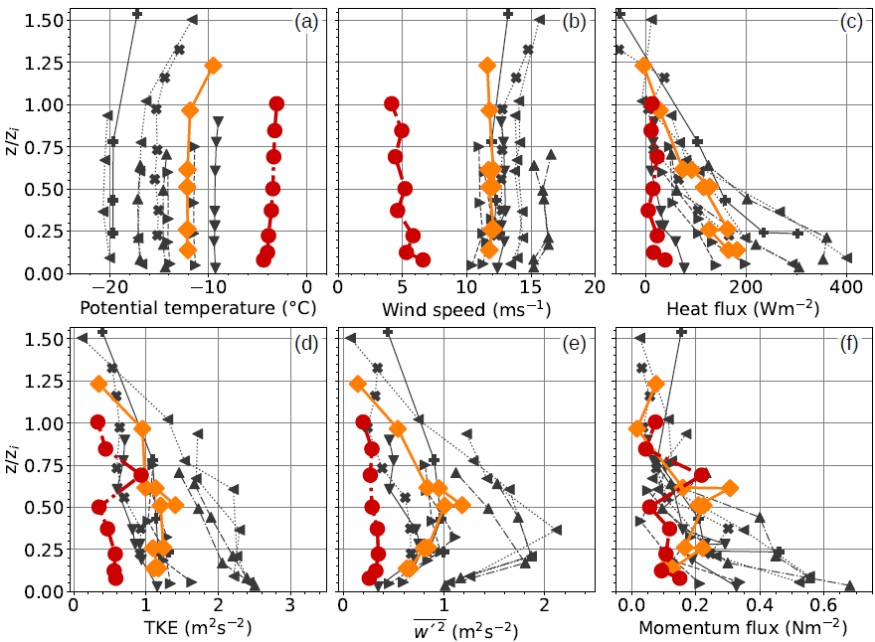

**Figure 20.** Comparison of atmospheric boundary layer (ABL) profiles, with $z$ representing altitude, of mean and turbulent quantities (nor-
malized by the ABL height $z_i$) obtained by airborne measurements during CAOs on 29 March (orange) and 9 April (red) 2022 during
HALO–$(\mathcal{AC})^3$. Data from previous aircraft campaigns (ARTIST, Hartmann et al. (1999), and AFLUX, Mech et al. (2022)) are indicated by
black lines with different symbols. TKE is the turbulent kinetic energy, and $\overline{w'^2}$ represents the vertical velocity variance.



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
