# Peer review of "Overview: Quasi-Lagrangian observations of Arctic air mass transformations – Introduction and initial results of the $HALO-(AC)^3$ aircraft campaign"

_EGUsphere, 2024_

## Referee Comment (RC2)

This manuscript gives an introduction to the quasi-Lagrangian sampling strategy performed during the HALO-(AC)3 campaign in order to study air mass transformations. Some example applications/first results are described, covering the observations during air mass transformation in warm-air intrusion (WAI) and cold-air outbreak (CAO) cases, Arctic cloud and aerosol properties, and measurements of mesoscale divergence and subsidence.

In general, it is a very well written manuscript, providing a good overview of the quasi-Lagrangian sampling strategy and its application. I do have a few specific/minor comments for improving the clarity in some places, apart from these, I believe that the manuscript will be publishable.

Comments:

line 60/footnote: Why do you restrict yourself to marine CAOs? Please explain and mention here.

ll 96-104: As you list previous attempts of Lagrangian measurements: What do these studies tell us here? Was their approach successful, did you copy their approach? Where there issues which you now tried to solve with your approach? What can we learn from these studies which is valuable here?
Listing these is good, but what do they tell us for this study? Is there an added value we can gain from these studies?

line 133: Ehrlich et al., 2024 – as this paper is not yet published, not even submitted, I would strongly recommend to make an according statement here: Ehrlich et al. (2024, to be submitted).

line 134: AWIPEV – abbreviation?

Table 1: Dropsondes used in GTS: Are these taken from the HALO dropsondes only, or is it a mix of HALO and P5 dropsondes?

Section 3.1: Calculation of trajectories: Were these trajectories calculated a) before the flight – used for detailed flight planning, b) during flight, using 'real' starting position of the planes – for real-time steering of the planes, or c) after flight – to check whether quasi-Lagrangian sampling has been achieved?
The question is partly answered in the following subsections, but I think it would be good to state this clearly here.

Figure 2: Is HALO's remote sensing view on the air mass parcels (blue cubes) partly blocked by the P5, as it is flying stacked in between HALO and the air parcel? Or has this been taken into account for the collocated flight planning (e.g. adding a small spatial offset)?

Figure 3: I would recommend to let the aircraft nose point into the direction of flight. E.g. in Fig. 3b it looks as if the aircraft is flying outbound of Kiruna, but I believe, it is on its way back, so the nose should better be pointing towards the right (or downwards right).

ll 266-268: Why treat open ocean and sea ice differently in CAO and WAI cases? Why disregard the cases above sea ice in the former?

Figure 7: There seem to be some white boxes overlaying some of the subfigures: "(a)" and "(b)" is only displayed partly, some of the y-axis annotation of subfigure (b) are partly gone, the indication of cases (e) in subfigure (a) and case (c) in subfigure (b) are only partial.

line 409: "thin ice of nilas" ? What is nilas?

Figure 13: I would prefer if the colour scales also mention what property/variable they are showing. Secondly, the colour scale between subfigure a/b and c looks like it would belong to subfigures a and b, but I believe it does belong to subfigure c? If so, please reorder the figure to make this more clear, maybe place the colour scale on the right hand side of subfigure c? Also, in the figure, I can only spot instances where the occurrences (?) reach about or a bit more than 1000, while the scale goes up to 10000. Is that necessary? Otherwise, I would recommend to not extend so far, then also differences in the 30 – 100 value range (green to yellow transition) would become more visible. For all three subfigures: please consider changing the colour table ("end the rainbow"): Generally, it is advised to choose colourblind-friendly colour schemes, and the rainbow scheme is unfortunately not one of those (among other shortcomings of this colour table, see e.g. the open letter to the scientific community here: https://www.climate-lab-book.ac.uk/2014/end-of-the-rainbow/).

lines 450-452: How can you be sure that the aerosol particles that were observed below and above the cloud are cloud droplet residuals (CDR)? Or do you mean that you compared the CDR properties (from within clouds) to aerosol properties above and below cloud? Please clarify/rephrase.

Figure 15: How were the size distributions normalised, in what respect?

Figure 17 b-e: While you indicate times for ERA5 in the figure labels, could you also do so for the HALO/P5 lines?
subfigure d) – The label indicates a green colour for the P5, in the plot, I can only find a red line, please check (maybe worth checking all figures again for colourblind-friendliness).

Appendices:
While I understand that these results might be very interesting, the link to the main paper is not clear to me. They get barely mentioned (just in the introduction of Section 4 (" In the three appendices, we add partly preliminary, but nonetheless very interesting supplementary discussion and results from the HALO–(AC)3 campaign"), but seem otherwise disconnected. As the manuscript is already very long, I am not convinced that these appendices are necessary. So, maybe consider removing these. What is their link to the main aim of the paper (quasi-lagrangian observations), and what value do they add in that regard?

Figure 20: A legend would be nice.

---

## Author Comment (AC1)

**Comments and Suggestions by REVIEWER 1**

**SUMMARY**

**Reviewer 1:** This paper provides an overview of the recent HALO-(AC)$^3$ aircraft campaign highlighting the flight plans/strategy, measurements, and various measurement/sampling techniques of Arctic air masses. Some of the novelties of this study include quasi-Lagrangian measurements from eight warm-air intrusion (WAI) and twelve cold-air outbreak (CAO) cases, derived surface heating/cooling and moistening/drying estimates, comprehensive aerosol/CCN data, and estimates of mesoscale divergence using dropsondes released during circular flight patterns. The abstract is very well written, concise, and clearly conveys the novelties and (initial) results of the AC$^3$ campaign. The quasi-lagrangian sampling strategy is very clearly defined, thought out, and easy to follow in the text/results. The section on Arctic clouds nicely highlights cloud phase as a function of the underlying surface (open water versus sea ice) as well as a simultaneous retrieval of effective radius for both ice crystals and liquid drops. These results, in my view, appropriately highlight and contextualize the various datasets as well as set the table for a number of planned (and likely interdisciplinary) analyses across a wide array of Arctic climate science sub-disciplines. Aside from a couple of very minor comments (indicated in the Specific Comments) and with a few of the figures being quite "busy" with lines and markers, every figure – in my view – is justified in its content with each figure adding very clear and rich context to the paper. Another strength of this manuscript is that, given volume of data and analysis in this manuscript, all sources of uncertainty (e.g., LWP and snowfall) are well characterized and quantified.

This is a commendable effort by all authors and contributors. For a very lengthy manuscript with 17 figures and 3 appendices, this was a very fun read with a lot of concise, "to-the-point" information that many sub-disciplines within the Arctic science community will be eager to read. I liken this manuscript to a fine 7-course dinner: it may take a while before you're finished, but every course delivers masterfully crafted dishes by world-class chefs with each dish delivering a palette of flavors certain to whet every appetite in the Arctic climate community. The manuscript in its present form is perfect in the sense that it captures just the right amount of detail (in my view) for an overview paper. While I have a number of very specific comments that would improve clarity in a few spots, they are extremely minor and can be addressed quickly without the need for a second review. I have no general concerns/comments for this manuscript, and overall, I believe this manuscript is publishable in its present form to Atmospheric Chemistry and Physics.

I look forward to many more in-depth studies following and building upon the excellent work presented in this manuscript.

**_Reply:_** _Thank you for your general comments, which we greatly appreciate. Also, thanks a lot for your very useful specific suggestions. We did our best to carefully consider all your remarks._

**SPECIFIC COMMENTS**

**Reviewer 1:** L23-24: "... was more than 1.5 K warmer than during pre-industrial times" though it's stated "Data published by the Copernicus Climate Change Service show...", this statement needs a citable reference.

***Reply:** We give the following web site as a reference for this statement: https://climate.copernicus.eu/global-climate-highlights-2023*

*The corresponding sentence has been changed, such that it is more precise now:*

*"The data published by the Copernicus Climate Change Service show that on almost 50 % of days in 2023, the anthropogenic warming exceeded the values of the pre-industrial period (1850-1900) by at least 1.5 K (https://climate.copernicus.eu/global-climate-highlights-2023)."*

**Reviewer 1:** L24: "numerous feedback mechanisms in the Earth's climate system" it would be good to list 2-3 or so of these feedback mechanisms here.

Reply: *We have added one sentence listing several Arctic-relevant feedback mechanisms here:*

*"Prominent examples of these Arctic-relevant feedback loops are the Planck, water vapor, surface albedo, and cloud effects."*

**Reviewer 1:** L44: A reference or two here would be good.

***Reply:** Yes, we have included two references to substantiate this statement:*

- *Alvarez, J., Yumashev, D., and Whiteman, G.: A framework for assessing the economic impacts of Arctic change, Ambio, **49**, 407–418, https://doi.org/10.1007/s13280-019-01211-z, 2020.*
- *Melia, N., Haines, K., and Hawkins, E.: Sea ice decline and 21st century trans-Arctic shipping routes, Geophysical Research Letters, 43, 9720–9728, https://doi.org/10.1002/2016gl069315, 2016.*

**Reviewer 1:** L105: This is a very lengthy introduction, but a necessary one as each paragraph here has a clear focus and motivation for the AC3 campaign.

***Reply:** We agree and have not changed this part.*

**Reviewer 1:** L124: Add latitude/longitude coordinates for Kiruna and Longyearbyen here.

*Reply: We have included the geographical coordinates of Longyearbyen (78.24° N, 15.49° E) and Kiruna (67.85° N, 20.22° E).*

**Reviewer 1:** L134: Add latitude/longitude coordinates for Ny-Ålesund.

*Reply: We have added the geographical coordinates of Ny- Ålesund: 78.92°N, 11.92°E.*

**Reviewer 1:** Section 2, like the introduction, is very well structured and written.

*Reply: Again, we agree.*

**Reviewer 1:** L171: Casual readers may not fully understand what a "Lagrangian" frame of reference is and how it ties into the sampling strategy described in this paragraph. A sentence to open up this paragraph describing what "Lagrangian" is, in my view, would lead the rest of this paragraph better and make the sampling strategy clearer to the reader in its objective.

*Reply: Thanks, we now introduced the term "Langrangian" even earlier, in the introduction, right after the first mention of the term. We have changed/added the text in the "Introduction" section as follows:*

*"As a consequence, dedicated observations of WAIs and CAOs would be helpful to improve the model capabilities in order to realistically represent processes that determine air mass transformations during meridional transports into and out of the Arctic (Wendisch et al. 2021). Lagrangian measurements are well suited for this purpose. The Lagrangian approach assumes that the observations are made in relation to a coordinate system that moves together with the air mass. In this way, the changes in the properties of the same air parcel can be observed along its pathway. In contrast, the observations from a Eulerian perspective refer to a locally fixed coordinate system, so that the properties of successive, different air parcels are measured from a fixed position as a time series."*

**Reviewer 1:** L174: "Because of their..." I would lead this sentence with "For example, ..." as this would more clearly lead the reader into a discussion of balloon-related drawbacks described in the previous sentence.

**Reply:** *Done.*

**Reviewer 1:** Figure 2 Caption: Recommend changing "enables to observe the changes" to "enables observational changes"

**Reply:** *We have replaced "enables to observe the changes" with "enables observing the changes". Otherwise we would not meet what we intend to say.*

**Reviewer 1:** L248: I am slightly confused by the writing here – what do you mean by a "quality of possibilities"? I think "provides unprecedented quantity of possibilities" would work here.

**Reply:** *Indeed, your suggestion makes perfect sense. Thanks, we have modified the text accordingly.*

**Reviewer 1:** Figure 5: This is a very well-constructed figure that clearly contrasts CAOs with WAIs.

**Reply:** *We agree. To make this even clearer we have modified the figure slightly, see below:*

[Figure]

**Reviewer 1:** L305: How exactly is the "ice growth process" inferred or done using measurements here?

**Reply:** *This has been realized by using in-situ measurements of ice crystal size distributions that are described in detail by Maherndl et al. (2024).*

**Reviewer 1:** L306: Can you point to or reference where "we also detect stronger riming"?

**Reply:** *Again, we refer to Maherndl et al. (2024) cited in our paper.*

**Reviewer 1:** Figure 6: Very picky comment here... "weak" should be capitalized in the Figure Title.

**Reply:** *Done. We appreciate that you are picky, because we are picky ourselves.*

**Reviewer 1:** L317-319: Very interesting result!

**Reply:** *We agree.*

**Reviewer 1:** Figure 7: I love the setup of this figure – it is definitely one of the most informative figures I've ever seen relating ice index and distance from the ice edge to actual cloud morphology. I hope to see versions of this figure in your future papers.

**Reply:** *We are working on more detailed papers on this topic.*

**Reviewer 1:** Figure 8 caption: Is it really necessary to call this a "Shapiro-Keyser cyclone" here? I think it would be better if this were referenced (including the citation) in the main text rather than the figure caption.

**Reply:** *We have shifted this part into the main text, following your advice.*

**Reviewer 1:** L374 and Figure 10 caption: One of the other prevailing cloud phase/microphysics algorithms for ground-based cloud remote sensors follows the widely-used Shupe et al. (2008, and references therein). I think it would be useful for the Arctic cloud/climate community to comment on how your algorithm compares with the Shupe et al. algorithm (and perhaps discuss

how a comparison of these algorithms might be done in a future AC3-related study which would also be very interesting!).

Shupe, M. D., and Coauthors, 2008: A Focus On Mixed-Phase Clouds. Bull. Amer. Meteor. Soc., 89, 1549–1562, https://doi.org/10.1175/2008BAMS2378.1.

*__Reply:__ In the in-situ cloud community, there is no universally applied definition of a mixed-phase cloud. Various methods, such as those described in Korolev et al. (2017), examine the ratio of liquid to ice in different ways and are strongly dependent on the in-situ cloud instruments used. The method we apply here has been validated using a Polar Nephelometer, which directly indicates the thermodynamic phase of cloud particles through their optical properties. This approach is particularly advantageous because particle sizing instruments normally require additional assumptions to differentiate between solid ice and liquid water.*

*The new method for thermodynamic cloud phase classification presented in Moser et al. (2023) can be used to develop new retrieval algorithms or to validate existing remote sensing retrievals for the detection of Arctic mixed-phase clouds, such as discussed in Shupe et al. (2008). Since many data from remote sensing instruments and in-situ cloud probes were obtained in colocation during the HALO-(AC)³ campaign, further studies on algorithms for the microphysical propertied of Arctic clouds will be investigated within the (AC)³ project.*

*To consider this issue raised by the reviewer we have added one sentence (including corresponding references) after line 385:*

*"Furthermore, the method to detect thermodynamic phase in Arctic mixed-phase clouds with in-situ particle measurements as describes in Moser et al. (2023) will be used to validate existing remote sensing algorithms, such as that of Shupe et al. (2008)."*

*References:*

- *Shupe et al., 2008: https://doi.org/10.1175/2008BAMS2378.1*
- *Korolev et al. 2017: https://doi.org/10.1175/AMSMONOGRAPHS-D-17-0001.1*
- *Moser et al. 2023: https://doi.org/10.5194/acp-23-7257-2023*

**Reviewer 1:** L382: Just say "Future studies" rather than "near future studies".

*__Reply:__ Done.*

**Reviewer 1:** L390: Following my previous comment for L374, this might be a good spot to discuss potential differences in these algorithms.

*__Reply:__ Done above, hopefully.*

**Reviewer 1:** L430: How typical are RHi values of 140%? Might be good to add a reference or two here for comparison sake.

*__Reply:__ High values of supersaturation (larger than 140 %) within cirrus clouds have also been reported by former studies (e.g., Comstock et al., 2004; Groß et al., 2014; Krämer et al., 2020). However, they did not focus on cirrus clouds in the Arctic. In contrast, Gierens et al. (2020) used radiosonde measurements for cirrus cloud studies in the Arctic and found high ice supersaturation; sometimes even exceeding 150%. Here are the respective references:*

- *Gierens, K. M., Wilhelm, L., Sommer, M., & Weaver, D. (2020). On ice supersaturation over the Arctic. Meteorologische Zeitschrift, 1-12.*
- *Krämer, M., Rolf, C., Spelten, N., Afchine, A., Fahey, D., Jensen, E., ... & Sourdeval, O. (2020). A microphysics guide to cirrus–Part 2: Climatologies of clouds and humidity from observations. Atmospheric Chemistry and Physics, 20(21), 12569-12608.*
- *Groß, S., Wirth, M., Schäfler, A., Fix, A., Kaufmann, S., & Voigt, C. (2014). Potential of airborne lidar measurements for cirrus cloud studies. Atmospheric Measurement Techniques Discussions, 7(4), 4033-4066.*
- *Comstock, J. M., Ackerman, T. P., & Turner, D. D. (2004). Evidence of high ice supersaturation in cirrus clouds using ARM Raman lidar measurements. Geophysical Research Letters, 31(11).*

*We have reformulated the corresponding sentence as follows and added the reference:*

*"Even values exceeding the threshold of homogenous freezing have been found inside and around the WAI cirrus. This is in accordance with former findings of Gierens et al. (2020), who used radiosonde measurements to study cirrus clouds in the Arctic."*

**Reviewer 1:** L455-456: I'd merge these two sentences.

*__Reply:__ Done.*

**Reviewer 1:** L512-513: I agree with this conclusion.

*__Reply:__ Thanks.*

---

## Author Comment (AC2)

**Comments and Suggestions by REVIEWER 2**

**Reviewer 2:** This manuscript gives an introduction to the quasi-Lagrangian sampling strategy performed during the HALO-(AC)3 campaign in order to study air mass transformations. Some example applications/first results are described, covering the observations during air mass transformation in warm-air intrusion (WAI) and cold-air outbreak (CAO) cases, Arctic cloud and aerosol properties, and measurements of mesoscale divergence and subsidence. In general, it is a very well written manuscript, providing a good overview of the quasi-Lagrangian sampling strategy and its application. I do have a few specific/minor comments for improving the clarity in some places, apart from these, I believe that the manuscript will be publishable.

*Reply: Thanks a lot for your very useful review. We did our best to carefully consider all your suggestions and remarks.*

**Reviewer 2:** line 60/footnote: Why do you restrict yourself to marine CAOs? Please explain and mention here.

*Reply: This restriction is determined by the $(AC)^3$ project, which excludes by purpose investigations related to land ice atmosphere interactions, for example in relation to Greenland. Instead, $(AC)^3$ is focusing on marine environments. We simply don't have the funding to cover the land masses in addition to the sea surface processes. To make this clear we have modified the footnote to:*

*"In this paper we restrict ourselves to marine CAOs; our project does not investigate land surfaces such as the Greenland ice sheets."*

**Reviewer 2:** ll 96-104: As you list previous attempts of Lagrangian measurements: What do these studies tell us here? Was their approach successful, did you copy their approach? Where there issues which you now tried to solve with your approach? What can we learn from these studies which is valuable here? Listing these is good, but what do they tell us for this study? Is there an added value we can gain from these studies?

*Reply: We were hesitant to claim that we are the first who have really successfully applied this approach during a major aircraft campaign, we leave this to judge by the readers. But in fact, there are only really few attempts of quasi-Lagrangian measurements applying aircraft and to our knowledge there are none in the Arctic yet. We have invested many efforts to realize this technique, and even during the campaign we could not be absolutely sure that we have succeeded. That became only evident after eventually analyzing the data after the campaign coming up with the results shown in Figure 4. The list and discussion of the previous attempts is important to give the reader an overview and to show the issues related to this approach. We do hope that our description will help the reader to evaluate the success of our quasi-Lagrangian technique that enables, for the first time, to quantitatively analyze the ability of models to represent air mass transformation during meridional transport into and out of the Arctic.*

**Reviewer 2:** line 133: Ehrlich et al., 2024 – as this paper is not yet published, not even submitted, I would strongly recommend making an according statement here: Ehrlich et al. (2024, to be submitted).

*Reply: We are confident to come up with a citable reference for the data paper soon enough before publication of this current manuscript. Meanwhile we follow your suggestion and cite the data paper with "Ehrlich et al. (2024) for Submission to Earth Syst. Sci. Data".*

**Reviewer 2:** line 134: AWIPEV – abbreviation?

*Reply: AWIPEV is actually not a classical abbreviation, but a mixture of the institute abbreviations (with an "I" falling by the wayside) and in this form the name of our station. For more explanation, we have replaced the corresponding sentence by:*

*"Furthermore, intensive ground-based measurements were carried out at the permanent German-French AWIPEV research base operated by German Alfred Wegener Institute (AWI) and the French Polar Institute Emile Victor (IPEV) at Ny-Ålesund (Svalbard), including additional observations with a tethered balloon (Lonardi et al., 2024)."*

**Reviewer 2:** Table 1: Dropsondes used in GTS: Are these taken from the HALO dropsondes only, or is it a mix of HALO and P5 dropsondes?

*Reply: Thanks for finding this missing but important detail. The direct transmission to GTS was setup for HALO only. Therefore, all sondes listed here for GTS are from HALO. We adjusted the table caption accordingly.*

**Reviewer 2:** Section 3.1: Calculation of trajectories: Were these trajectories calculated a) before the flight – used for detailed flight planning, b) during flight, using 'real' starting position of the planes – for real-time steering of the planes, or c) after flight – to check whether quasi-Lagrangian sampling has been achieved? The question is partly answered in the following subsections, but I think it would be good to state this clearly here.

*Reply: This question is clearly answered at the beginning of Section 3.2 where it belongs to from our point of view. There we write:*

*"The actual computation of the forward-trajectories of the air mass parcels was performed using the Lagrangian analysis tool (LAGRANTO) (Sprenger et al. 2015). During the campaign, the trajectory calculations were based on the Integrated Forecast System (IFS) of the European Centre for Medium-Range Weather Forecasts (ECMWF) wind product. For processing the data after the campaign (for this paper) we have applied the Fifth Generation ECMWF Atmospheric Reanalysis (ERA5) (Hersbach et al. 2020)."*

*We would be hesitant to move that information to section 3.1. That might cause some confusion. We hope we could convince the reviewer.*

**Reviewer 2:** Figure 2: Is HALO's remote sensing view on the air mass parcels (blue cubes) partly blocked by the P5, as it is flying stacked in between HALO and the air parcel? Or has this been taken into account for the collocated flight planning (e.g. adding a small spatial offset)?

***Reply:*** *HALO and P5 have quite different flight speeds. Therefore we planned (or better we tried to plan) that both aircraft meet for a split of seconds somewhere in the middle of a flight leg. Anyway, even if the two aircraft would fly exactly above each other with the same speed, the influence of P5 on HALO remote sensing measurements in a distance between 8-10 km can safely be neglected.*

**Reviewer 2:** Figure 3: I would recommend to let the aircraft nose point into the direction of flight. E.g. in Fig. 3b it looks as if the aircraft is flying outbound of Kiruna, but I believe, it is on its way back, so the nose should better be pointing towards the right (or downwards right).

***Reply:*** *Thanks for that hint; we have modified Figure 3 correspondingly. In addition, we have removed Figure 3c and instead implemented two new panels to increase the resolution of the time series and to avoid blank spaces. Here is the new version of this figure considering your comment:*

[Figure]

**Reviewer 2:** ll 266-268: Why treat open ocean and sea ice differently in CAO and WAI cases? Why disregard the cases above sea ice in the former?

**_Reply:_** _The air mass transformations governing WAIs and CAOs are dominated by different processes. In the case of CAOs, the major driver are the surface turbulent heat fluxes arising from the pronounced temperature and humidity gradient between surface and near-surface air layers over open ocean (1,2,3, references see below). This results in the strongest diabatic heating and moisture uptake of near-surface air layers in the first few hours of CAO evolution (4). While the pre-conditioning of Arctic air masses over closed sea ice through continuous diabatic cooling is crucial for the CAO formation, the air temperature change rates are typically one order of magnitude smaller over sea ice than later over open ocean (2,4)._

_Contrary to CAOs, the processes driving air mass transformations in WAIs are typically more diverse. WAIs have been categorized into turbulence-dominated vs. radiation-dominated (5), and both cases are further complicated in the presence of a marked polar dome (5,6). Turbulent processes within WAIs are modified not only by the high air temperature and humidity of the lower-latitude as compared to the surrounding local air masses, but also the characteristic presence of thick, often stacked cloud layers (7,8,9,10). Thus, air mass tranformations in WAI flows during poleward transport can take place both over open ocean, the marginal sea ice zone, as well as over sea ice, and we would not want to miss the changes happening over all those different surface types._

_We have added the following sentences to the manuscript elaborating this procedure of treating open ocean and sea ice differently in CAO and WAI cases.:_

_"In CAOs, major air mass transformations occur over the open ocean due to intense surface turbulent heat fluxes driven by temperature and humidity gradients, whereas the preconditioning over Arctic sea ice typically involves rates one order of magnitude smaller (2,4). Thus, in this article we focus only on the processes setting in over open ocean in case of CAOs. On the contrary, during WAIs intense air mass transformations through turbulent, radiative, and cloud processes can set in over open ocean, the marginal sea ice zone, as well as the sea ice (7, 8, 5). Therefore, we do not restrain the analysis of air temperature and moisture changes during WAIs to any surface type."_

_References:_

   (1) _Brümmer et al. 1996 (10.1007/BF00119014)_
   (2) _Papritz and Spengler 2017 (0.1175/JCLI-D-16-0605.1)_
   (3) _Dahlke et al. 2022 (10.1029/2021JD035741)_
   (4) _Kirbus et al. 2024 (10.5194/acp-24-3883-2024)_
   (5) _You et al. 2022 (10.5194/acp-22-8037-2022)_
   (6) _Komatsu et al. 2018 (10.1038/s41598-018-21159-6)_
   (7) _Woods et al. 2016 (10.1175/jcli-d-15-0773.1)_
   (8) _Johansson et al. 2017 (10.1002/2017gl072687)_
   (9) _Kirbus et al. 2023 (10.3389/feart.2023.1147848)_
   (10) _Dekoutsidis et al. 2024 (10.5194/acp-24-5971-2024)_

**Reviewer 2:** Figure 7: There seem to be some white boxes overlaying some of the subfigures: "(a)" and "(b)" is only displayed partly, some of the y-axis annotation of subfigure (b) are partly gone, the indication of cases (e) in subfigure (a) and case (c) in subfigure (b) are only partial.

*__Reply:__ There seems to be a technical issue, the figure we have included in the paper is copied here.*

[Figure]

**Reviewer 2:** line 409: "thin ice of nilas"? What is nilas?

*__Reply:__ Nilas is a common term that comprises the "young sea ice of a few centimeters thickness". To make this clear we have replaced the corresponding original sentence by an extended version, which reads:*

*"Beyond 40 km distance, the emitted radiation is governed by the surface, which is characterized by a mixture of pack ice and leads with relatively warm open water and young sea ice of a few centimeters thickness (nilas), whose surface is also warmer than the surfaces of pack ice and cirrus. Thus, we see an increase of the emitted upward radiance in this region."*

**Reviewer 2:** Figure 13: I would prefer if the colour scales also mention what property/variable they are showing. Secondly, the colour scale between subfigure a/b and c looks like it would belong to subfigures a and b, but I believe it does belong to subfigure c? If so, please reorder the figure to make this clearer; maybe place the colour scale on the right hand side of subfigure c? Also, in the figure, I can only spot instances where the occurrences (?) reach about or a bit more than 1000, while the scale goes up to 10000. Is that necessary? Otherwise, I would recommend to not extend so far, then also differences in the 30 – 100 value range (green to yellow transition)

would become more visible. For all three subfigures: please consider changing the colour table ("end the rainbow"): Generally, it is advised to choose colour blind-friendly colour schemes, and the rainbow scheme is unfortunately not one of those (among other shortcomings of this colour table, see e.g. the open letter to the scientific community here: https://www.climate-lab-book.ac.uk/2014/end-of-the-rainbow/).

*__Reply:__ Thanks for this comment; we have revised the figure correspondingly. We changed color table and rearranged the Figure but we had to keep the large values in the right scale. This is an automatic scale adapted to the maximum values (even if their occurrence is little). Here is the adapted figure:*

[Figure]

**Figure 13.** Cross-section of (a) backscatter ratio at 532 nm, and (b) relative humidity with respect to ice (RHi) for RF03 on 13 March 2022 during a WAI. (c) shows a histogram of joint occurrence of RHi and backscatter ratio at 532 nm. The relative humidity was calculated from WALES water vapor measurements and model temperature field.

**Reviewer 2:** lines 450-452: How can you be sure that the aerosol particles that were observed below and above the cloud are cloud droplet residuals (CDR)? Or do you mean that you compared the CDR properties (from within clouds) to aerosol properties above and below cloud? Please clarify/rephrase.

*__Reply:__ The Reviewer is correct, the sentence has been revised to "We have compared the CDR aerosol properties __to those of ambient particles__ collected in the ABL (below cloud) and in the free troposphere (above cloud)".*

**Reviewer 2:** Figure 15: How were the size distributions normalised, in what respect?

*__Reply:__ Again, we would like to thank the Reviewer for his careful comments. We have added the missing information about the normalization in the caption of Figure 15: We have replaced*

*"Normalized number size distribution of cloud droplet residuals (CDR) ..." by*

*"Number size distributions normalized by the respective number concentration of cloud droplet residuals (CDR) ..."*

**Reviewer 2:** Figure 17 b-e: While you indicate times for ERA5 in the figure labels, could you also do so for the HALO/P5 lines? subfigure d) – The label indicates a green colour for the P5, in the plot, I can only find a red line, please check (maybe worth checking all figures again for colourblind-friendliness).

*__Reply:__ In the revised figure we have now indicated times for ERA5 and HALO/P5 in the figure labels, thanks for this hint. The label for P5 in panel d is now in red, as indicated in the caption. Here is the revised version of the figure:*

[Figure]

**Figure 17.** (a) Overview of mesoscale flight patterns flown by HALO (green circles) and P5 (red square pattern) sampling a low-level air mass on 29 and 30 March 2022 at four locations along its southbound trajectory (yellow lines). Dropsonde launch locations are marked by crosses (green if dropped by HALO, red crosses for sondes dropped by P5). (b)-(e) Profiles of pressure velocity Ω calculated from dropsonde data (time of first and last dropsonde launch indicated) released from HALO and P5 (solid lines) and ERA5 reanalysis data (dashed).

**Reviewer 2:** Appendices: While I understand that these results might be very interesting, the link to the main paper is not clear to me. They get barely mentioned (just in the introduction of Section 4 ("In the three appendices, we add partly preliminary, but nonetheless very interesting supplementary discussion and results from the HALO–(AC)3 campaign"), but seem otherwise disconnected. As the manuscript is already very long, I am not convinced that these appendices are necessary. So, maybe consider removing these. What is their link to the main aim of the paper (quasi-lagrangian observations), and what value do they add in that regard?

*__Reply__: We agree and have deleted the Appendix. Instead, we have included some sentences in the outlook of the manuscript. They read as follows:*

- *We will investigate the hypothesis that a secondary circulation acts to spread out moisture from the initial river-like intrusion in the cross-flow direction. This hypothesized circulation consists of uplift in the core of the intrusions, divergence in the upper troposphere above the core, and convergence and subsidence of drier air on the flanks of the intrusion. We will investigate the typical structure of the atmospheric moisture field, and how it evolves during a WAI including a discussion on what processes drive this development.*
- *The particular mode structure of the Arctic radiant energy budget (REB) will be analyzed as a function of altitude, different surface type (sea ice or open ocean), cloudy or cloud-free conditions, and thermodynamic properties (temperature lapse rate, horizontal temperature gradient between sea ice and open ocean). The observations will be confronted with model results that often struggle to correctly represent the mode structure of the REB, e.g., due to limitations in the treatment of sub-grid processes including clouds and the sea ice albedo. Therefore, detailed observation–model comparisons are envisioned to identify potential misrepresentations of properties affecting the REB.*
- *We are about to study an extraordinary CAO case observed during HALO-(AC)³. We will integrate our data obtained during two CAO events with measurements following approximately a North-South trajectory (29 March and 9 April 2022) with literature data. The 9 April case was extraordinary for two reasons. Firstly, it was characterized by cloud-free conditions along the entire 180 km North-South flight track across the MIZ to the open ocean, which is very rare and unusual. Only at the southernmost position, convective clouds appeared. Secondly, the southernmost position was influenced by the front of a polar low, which was a remnant of a polar low over Fram Strait on the preceding day. We will discuss how the results of our observations on these two dates fit to earlier measurements.*

**Reviewer 2:** Figure 20: A legend would be nice.

*__Reply:__ Again, this seems to be a technical problem. However, because the Appendix has been removed, this figure is not included in the manuscript anyways.*